# Insights From a Data- and Space-Agnostic Approach to Zero-Cost Proxies

**Timotée Ly-Manson**                                          *timotee.ly.manson@gmail.com*
*IMT Atlantique*
*UMR CNRS 6285 Lab-STICC*

**Mathieu Léonardon**                                    *mathieu.leonardon@imt-atlantique.fr*
*IMT Atlantique*
*UMR CNRS 6285 Lab-STICC*

**Abdeldjalil Aissa El Bey**                       *abdeldjalil.aissaelbey@imt-atlantique.fr*
*IMT Atlantique*
*UMR CNRS 6285 Lab-STICC*

**Reviewed on OpenReview:** *https://openreview.net/forum?id=sVWJczov4Q*

## Abstract

Zero-cost proxies (ZCPs) have enabled low-cost Neural Architecture Search (NAS) by removing the computational overhead from model training. However, important drawbacks of currently designed ZCPs remain unaddressed. While there is a strong correlation between ZCPs and model performance at the scale of entire search spaces, this does not necessarily translate to guiding the search to top-performing architectures. In this paper, we conduct extensive benchmarking over state-of-the-art proxies in the NAS-Bench-Suite-Zero setting and observe that the correlation decreases dramatically when reducing the space to the best architectures, demonstrating the presence of a top-rank gap. Moreover, embedded priors on search space and data make ZCPs unreliable across diverse tasks. We leverage adaptive parameter distribution statistics as a discriminator metric in the genetic framework and introduce ParaDis, a low-cost NAS algorithm that remains orthogonal to ZCP design, with potential to define a fully data- and space-agnostic search when paired with the right metric. Experiments on multiple benchmarks confirm that ParaDis reduces the top-rank gap across diverse tasks. ParaDis also achieves a test accuracy of $97.29 \pm 0.07$ % on CIFAR10 in the DARTS space, within 0.25% of state-of-the-art, remaining competitive against methods with heavier priors.

## 1 Introduction

Neural Architecture Search (NAS) has received increasing attention as it enables the design of neural network architectures without a human expert, but has long struggled to reduce its computational cost. Various methods introduced in the early days of NAS (Zoph & Le, 2017; Liu et al., 2018) treat the performance of the candidate architectures as a label to train a predictor. These remain the gold standard for finding the best architectures (White et al., 2021; Schrodi et al., 2023; Lin & Luo, 2026), but are impractical in real-world applications due to the large computational overhead of training each candidate from scratch.

Several lines of work related to reducing the computational cost of NAS have emerged, such as one-shot NAS (Pham et al., 2018; Guo et al., 2020; Chu et al., 2021; Liu et al., 2019) and zero-shot NAS (Mellor et al., 2021; Abdelfattah et al., 2021). In the latter, the training of candidate architectures is entirely foregone in

favor of ad-hoc metrics — so-called zero-cost proxies (ZCPs) — that estimate the performance of the model in its freshly initialized state. Methods that incorporate these proxies have achieved competitive results (Jiang et al., 2023; Peng et al., 2024) while reducing the computational load by several orders of magnitude.

Since ZCPs are integrated into various search routines (such as pruning-based or evolutionary algorithms), direct comparisons of final accuracy can be misleading. Consequently, Spearman correlation benchmarking on diverse tasks has become the standard to unify ZCP research and evaluate new metrics. In this work, we conduct thorough supplementary experiments within the NAS-Bench-Suite-Zero benchmark (Krishnakumar et al., 2022). By considering the benchmarks through the lens of the top ranks, we find that most ZCPs collapse at the most critical point of the NAS process: finding the best architecture in the space.

Prior work (Kadlecová et al., 2024; Krishnakumar et al., 2022) suggests that proxies are highly correlated with simpler metrics such as parameter count. A concrete example can be found by conducting an exhaustive search of the NAS-Bench-201 search space (Dong & Yang, 2020) on the CIFAR10 task (Krizhevsky et al., 2009) using state-of-the-art ZCP MeCo (Jiang et al., 2023). The top ranked architecture is composed solely of convolutions, indicating a strong link with parameter count at the top level, while gains of the method may originate from better ordering of mid-range performance architectures.

Additionally, zero-cost methods are often implemented with high reliance on space knowledge and task data. Search pipelines such as operation pruning (Chen et al., 2021a) rely on search spaces retaining a cell-like structure, while other attempts at enhancing the ZCP-driven genetic famework (Peng et al., 2024) rely on knowledge of search space statistics. ZCPs such as `zen` (Lin et al., 2021) presume the presence of specific architectural properties. On the other hand, zero-cost proxies are often computed using limited amounts of forward and backward passes, creating a dependency on data despite the aim to quantify inherent properties of the models. Some exceptions include the `synflow` metric (Abdelfattah et al., 2021). Space- and data-agnosticity is an often overlooked trait in ZCPs, despite being instrumental in ensuring the results of zero-cost methods can transfer to new spaces and diverse tasks.

In this work, we explore the potential of regularizing ZCP-driven exploration with parameter distribution statistics, building upon literature that identifies parameter count as a robust performance baseline (White et al., 2022; Javaheripi et al., 2022). Specifically, we introduce **Para**meter **Dis**tribution Shift NAS (**ParaDis**), an evolutionary-based framework that assigns ZCPs with a role of exploration while applying parameter-based space constriction in a fully search space-agnostic way. Through thorough evaluation of our method under the NAS-Bench-Suite-Zero (Krishnakumar et al., 2022) and DARTS (Liu et al., 2019) settings, we find that our method boosts the ability of all ZCPs to find better architectures among the top ranks despite the simplicity of the framework. Furthermore, the robustness of the ParaDis framework allows for a fully data-agnostic search configuration. By pairing ParaDis with a data-agnostic ZCP like `synflow`, we demonstrate that structural priors alone, when properly constrained, can yield results competitive with methods that rely on access to labeled data.

Our contributions are as follows:

1. We benchmark existing zero-cost proxies using the NAS-Bench-Suite-Zero (Krishnakumar et al., 2022) to test the limits of ZCP performance. Specifically, we demonstrate that while state-of-the-art proxies correlate well globally, their performance collapses when ranking high-performing architectures. We identify this phenomenon as the **top rank gap**.

2. We introduce ParaDis, a genetic-based NAS method that couples ZCP-guided exploration with a dynamic parameter distribution shift, constraining the search to high-potential regions of the space. ParaDis is orthogonal to ZCP design and makes no assumption about the search space.

3. We demonstrate through rank experiments on NAS-Bench-Suite-Zero that ParaDis can enhance the quality of architectures found by ZCPs, thus bridging the top rank gap.

4. We conduct experiments on the standard DARTS (Liu et al., 2019) search space and find that even under fully data- and space-agnostic constraints, ParaDis remains competitive with state-of-the-art ZCP-based methods.

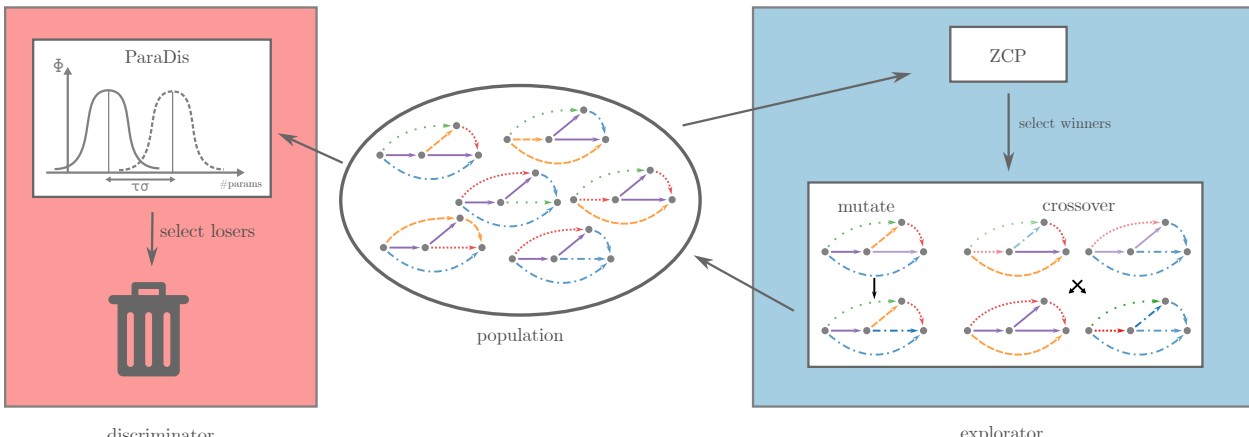

Figure 1: Our ParaDis method is based on the classic genetic zero-cost framework. Any ZCP can be used in the role of the explorer and select architectures that go on to reproduction. The ParaDis discriminator discards architectures based on parameter count distribution statistics over the population, with the size of architectures sampled at each generation controlled by $\tau$.

## 2 Related works

**Zero-cost proxies**   Zero-cost proxies (ZCPs) (Mellor et al., 2021; Abdelfattah et al., 2021) hold the potential to conduct neural architecture search without the need to train architectures. ZCPs are generally derived from mathematical properties of neural networks computed from their parameters or gradients, which can be obtained during a single forward or backward pass. While most ZCPs rely on actual data from the task, metrics such as `synflow` operate in a data-agnostic regime, and quantifiable properties of the networks such as parameter count (`#params`) can also be considered data-agnostic ZCPs. Compared to data-reliant ZCPs, data agnosticity yields better robustness when confronted with diverse tasks, and ease of implementation as a data loading pipeline is not required.

Various ZCPs have been introduced for the purpose of architecture search. Initially, Mellor et al. (2021) proposed to use the Hamming distance of the binary codes of the activations, a proxy which was later renamed `jacov`. Abdelfattah et al. (2021) brought metrics from the network pruning literature into the NAS space and coined the term ZCP. ZCPs tend to fall into two categories based on whether they aim to quantify the trainability of the candidate — that is, how easily it can converge to its optimal weights — or its generalization capabilities. Thus, the metrics `gradnorm`, `snip`, `grasp` and `synflow` are of the trainability type, while `fisher` is of the generalization type. This observation was further explored in Chen et al. (2021a), who paired the trainability measure `ntk`, based on the NTK condition number (Xiao et al., 2019), and the expressivity measure `#lr`, a count of the number of linear regions of the input space partitioned by the network (Raghu et al., 2017).

Recently, more advanced ZCPs have emerged and demonstrated competitive results on vision tasks. `zen` (Lin et al., 2021) leverages the statistics of the batch norm layers. `zico` (Li et al., 2023) exploits the link between the statistics of the gradient norms and network generalization ability. `meco` (Jiang et al., 2023) estimates the convergence rate via the minimum eigenvalue of the correlation matrix. `swap` (Peng et al., 2024) reframes the question of linear regions to create a more insightful metric.

**Zero-shot NAS**   ZCPs must be paired with a search strategy to explore the search space. Several possibilities have been explored in previous work. Mellor et al. (2021) proposed the naive random search solution, and warm-started other traditional NAS methods with the ZCP. Similarly, Abdelfattah et al. (2021) envisioned ZCPs as a warmup for other algorithms. Chen et al. (2021a) and Jiang et al. (2023) implemented a pruning-like strategy where a directed acyclic graph containing all possible architectural choices is pruned down to only the most desirable operation on each edge. More recently, Ji et al. (2025) leveraged ZCPs as context to drive architecture search by iteratively prompting a large language model.

However, the most popular search strategy in conjunction with ZCPs remains by far the genetic approach (Lin et al., 2021; Li et al., 2023; Peng et al., 2024). It is also a very common strategy for NAS in general (Real et al., 2019; Ericsson et al., 2024). In this work, we also adopt the evolutionary framework. While other NAS methods keep the algorithmic design to a bare minimum, choosing to instead focus on the proxies' capabilities, we directly improve on the classic genetic algorithm to cover the limits of ZCPs. Importantly, this allows us to remain orthogonal to ZCP design.

**ZCP ensembling**  Intuitively, using more than one proxy should be beneficial, since metrics that evaluate different aspects of a network might cover each other's blind spots (Lukasik et al., 2025). Chen et al. (2021a) first introduced this idea by summing their `ntk` and `#lr` metrics. He et al. (2024) generalized the idea by ensembling 6 ZCPs through weighted linear combination and learning the weights through Bayesian optimization. However, direct summation of ZCPs is impractical as the statistics of individual proxies are task-dependent and largely unknown. Without robust normalization, such sums are highly sensitive to outliers. A more robust approach can be found in AZ-NAS (Lee & Ham, 2024) which non-linearly aggregates ZCP scores as a single fitness score for genetic evolution.

ZCPs may also be used as inputs for supervised performance prediction. Kadlecová et al. (2024) ensemble proxies by treating them as features of the candidate architectures and applying random forest regression. Akhauri & Abdelfattah (2024) used ZCPs to train a surrogate among a multitude of other encodings. Similarly, Dong et al. (2025) used node-wise ZCP measures as inputs for the training of a surrogate fitted directly on Kendall-*tau* scores. More recently, Qin et al. (2025) fine-tuned language models on graph properties and ZCP measurements to evaluate architectures in high expressivity search spaces.

Differently, we assign roles to ZCPs within the genetic framework such that they guide exploration of the search space or discriminate architectures to constrain the space. As a result, ZCP aggregation methods such as AZ-NAS (Lee & Ham, 2024) can directly be paired with our ParaDis.

**Parameter count**  Simply counting the parameters of a candidate architecture is a naive, yet effective proxy for accuracy. Indeed, network complexity and generalization abilities grow with network size. Notably, the surprising effectiveness of the parameter count proxy `#params` often challenges ZCPs (White et al., 2022; Javaheripi et al., 2022), which are shown to be inferior to merely counting the parameters on various tasks. While this metric lacks the nuance to discern common architectural choices such as skip connections, a staple of architecture design shown to improve trainability (He et al., 2015; Orhan & Pitkow, 2018), complex ZCPs might identify these intricacies while failing to describe the bigger picture.

We propose that the parameter count may serve as a prerequisite metric and adopt it as a regularizer of the search. Peng et al. (2024) also used parameter count to regularize the score of their metric `swap` and focus the search around a desired model size. However, this requires prior knowledge of the search space's statistics, which is impractical for unseen spaces where the location of good architectures in the distribution is unknown. We instead design our method in a space-agnostic way by estimating parameter count statistics directly from available sampled architectures, removing the need for priors over the space. This allows the method to continue performing in real-world spaces with unknown structures.

## 3 Investigating the limitations of ZCPs

### 3.1 Context and benchmark description

Zero-cost proxies are benchmarked by evaluating their correlation to common search objectives such as validation accuracy. A number of classic ZCPs have been benchmarked by Krishnakumar et al. (2022) on a collection of 5 search spaces (Ying et al., 2019; Dong & Yang, 2020; Siems et al., 2020; Duan et al., 2021). In this work, we consider only 4 of these search spaces:

- NAS-Bench-201 (Dong & Yang, 2020) (NB201), a cell-based space of 15625 architectures for image classification trained on CIFAR-10, CIFAR-100 and ImageNet16-120.

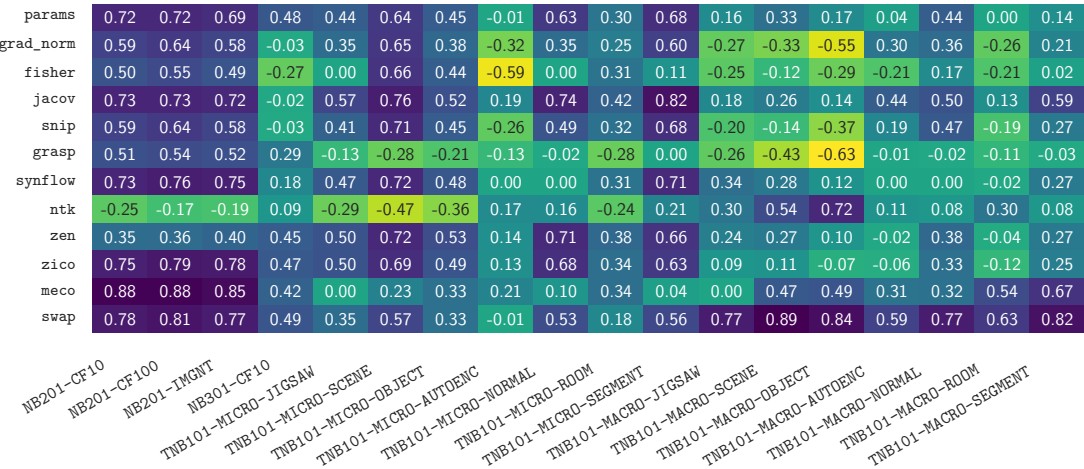

Figure 2: Spearman rank correlations of various ZCPs on the NAS-Bench-201, NAS-Bench-301 and TransNAS-Bench-101 benchmarks. Spearman was computed over 3 seeds against the entire search space.

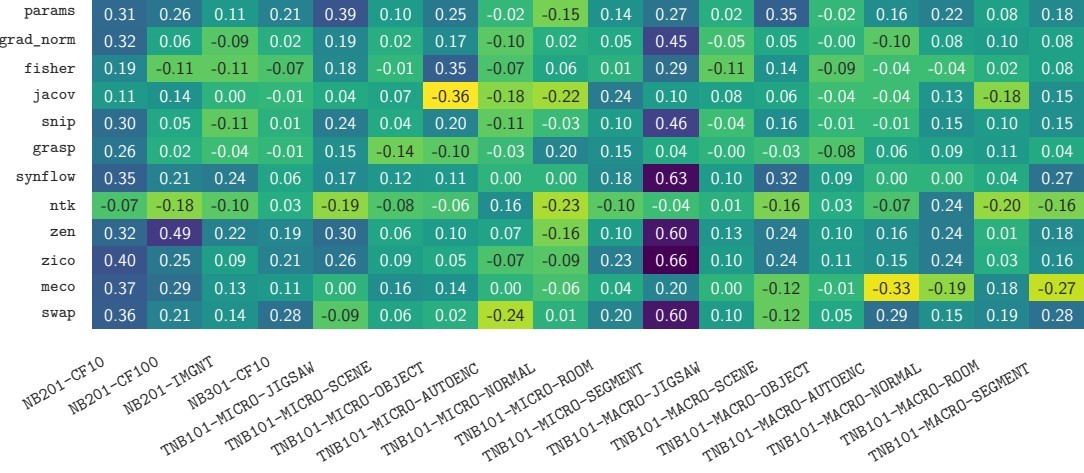

Figure 3: Spearman rank correlations of various ZCPs on the NB201, NB301 and TNB101 benchmarks. Spearman was computed on 3 seeds against the top 1% architectures in the space.

- NAS-Bench-301 (Siems et al., 2020) (NB301), a search space based on DARTS (Liu et al., 2019) and evaluated on CIFAR-10 via the surrogate method.

- TransNAS-Bench-101 Micro (Duan et al., 2021) (TNB101-Micro), a cell-based search space of 4096 architectures tailored for diverse tasks such as object detection in the Taskonomy task bank (Zamir et al., 2018) or autoencoding tasks (Krishnakumar et al., 2022).

- TransNAS-Bench-101 Macro (Duan et al., 2021) (TNB101-Macro), a similar space to TNB101-Micro but focused on the macro structure with a stackable cell in 3256 different configurations.

NB201 and NB301 are designed for image classification and compatible with the common CIFAR-10, CIFAR-100 and ImageNet16-120 tasks (Krizhevsky et al., 2009; Deng et al., 2009; Chrabaszcz et al., 2017), the TNB101 benchmark focuses on other paradigms and evaluating the performance of NAS across domains and tasks and outside of the classical cell-based framework. The metric used for the benchmark is the Spearman

rank correlation which is the correlation of the ranks of the ZCP scores $X$ and the ground truth scores $Y$:

$$r_s = \frac{cov(rank(X), rank(Y))}{\sigma_{rank(X)}\sigma_{rank(Y)}}$$

It indicates the average difference between the true rank of the candidate architecture and the rank given by the proxy. Proxies with high Spearman rank correlation tend to keep candidates ranked close to their true rank, such that a candidate is easily distinguished from others that it outclasses by far.

### 3.2 Benchmarking: changing the perspective

We first reproduce the benchmark proposed by Krishnakumar et al. (2022) while including several state-of-the-art proxies that have been proposed since the initial benchmark. The results are shown in Fig. 2. The NAS-Bench-101 benchmark is omitted due to its large size. As we can see, newer proxies, namely `meco` and `swap`, reach high correlation on a variety of different tasks.

However, the represented search spaces are very wide and heterogeneous. We observe that on many tasks, the distributions of architecture performance in the TNB101 search spaces form distinct clusters. Therefore, high correlation over the entire search space only indicates that the proxy can eliminate all weak architectures and reach a high-performance cluster. Overall, the benchmark does not answer the central question of Neural Architecture Search: how easily can we reach the top architecture in the space?

In order to obtain deeper understanding of ZCP behavior, we adopt a novel perspective on benchmarking. For each search space, we run the benchmark restricted to only the top 1% of architectures in the space, in terms of validation accuracy. Assuming that proxies are good enough to guide the search to the top 1% cluster in the space, we want to assess whether they can also distinguish the performance variations in that group. The results of the benchmark are shown in Fig. 3. We notice a significant drop in correlation. On most tasks, none of the proxies have a correlation higher than 0.5, less than half of the correlation over the whole space. Some metrics, such as `meco` and `jacov`, show no correlation or negative correlation with the top architectures, despite having positive correlation with the entire space. Moreover, we observe that the ranking of metrics is shuffled around as metrics such as `zen` seem better suited to the top 1% benchmark than state-of-the-art proxies for the overall benchmark, such as `meco` and `swap`.

This analysis shows that while zero-cost proxies have demonstrated the ability to weed out poorly performing architectures during the search, they cannot properly rank architectures after having reached a sub-space that contains only good candidates. Thus, considering the current state-of-the-art, there is a ceiling to the quality of architectures obtained through ZCP-driven search. We refer to this issue as the **top rank gap**.

Anterior works (Kadlecová et al., 2024; Krishnakumar et al., 2022) have pointed out that ZCPs are inherently biased towards easily identifiable characteristics of neural networks, such as the number of convolutions. Similarly, we compute the per-task correlation of all ZCPs against the parameter count of the candidate networks in Figure 4. We find that most ZCP rankings display $> 0.5$ correlation with parameter counts ranking. It is a well-known result in the literature (White et al., 2022; Javaheripi et al., 2022) that the parameter count of the network is a very strong proxy. Intuitively, more parameters in the network directly translate to enhanced expressivity. However, the complex search spaces of NAS present the additional challenge of connectivity affecting performance in ways that do not reflect in the parameter count. We hypothesize that ZCPs yield valuable information about network connectivity that is hard to exploit due to the overwhelming prior on parameter count. Therefore, including parameter count in the search process is a promising direction to make the most of ZCPs and overcome the performance ceiling.

## 4 ParaDis: Parameter Distribution Shifted NAS

We introduce **Para**meter **Dis**tribution Shift NAS, leveraging the genetic framework to conduct the search with a pair of metrics. For the basic genetic setup, we take inspiration from anterior elitist evolution algorithm proposals for NAS (Real et al., 2019; Peng et al., 2024).

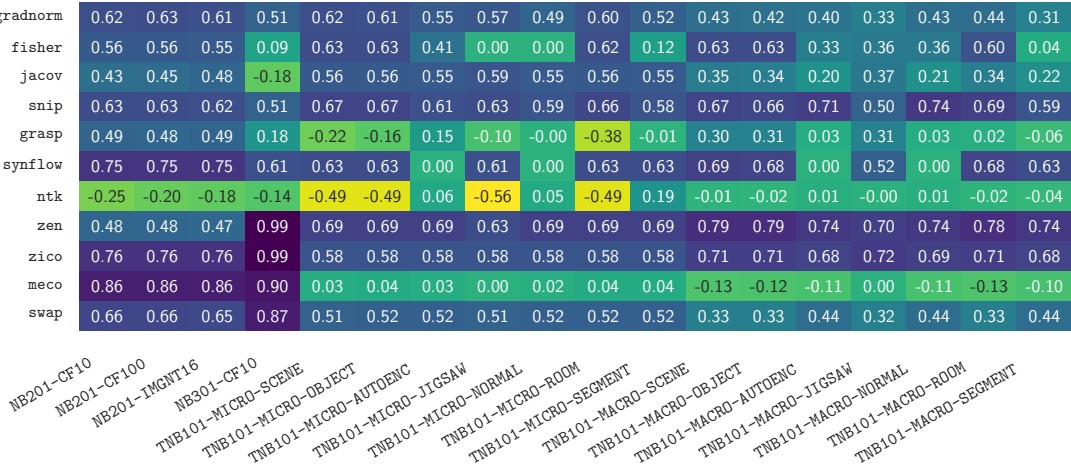

Figure 4: Spearman rank correlations of various ZCPs with `#params` on the NB201, NB301 and TNB101 benchmarks. Spearman was computed on 3 seeds against the entire search space.

Candidate architectures are encoded into a genome in the form of a sequence of numbers, each corresponding to a possible architectural choice in the search space. While the genomes are processed and interpreted differently depending on the definition of the search space, they are processed identically within the genetic algorithm, thus decoupling search space design from the search algorithm. The various genome encodings for each search space are described in Appendix C.

We allow pairs of metrics to drive the search by assigning them two distinct roles:

- **Explorer** metric: directs exploration of the search space by choosing evolutionary winners, which go on to mutation and crossover. This should typically be a metric that can distinguish good architectures. By design, ZCPs are well-suited to search space exploration.

- **Discriminator** metric: constrains the space by choosing evolutionary losers, which are removed from the population. This is a broad and reliable metric that identifies architectures to weed out in order to restrict the space to a desirable region. For this role, we define an adaptive parameter count distribution estimator, effectively constraining the space around the most suitable model size.

While we define these roles in the context of genetic algorithm design, the main intuition — that some metrics are better suited to choose good architectures, and others to detect bad architectures — holds across other NAS paradigms.

Additionally, this framework is easily generalizable to the regular zero-cost setup by assuming the explorer and discriminator to be the same. However, most zero-cost proxies are designed for high-scoring architectures and are therefore more suited for the exploration role. For this reason, the explorer is flexible and can be any zero-cost proxy. This keeps the method fully orthogonal to proxy design. Importantly, other means of aggregating ZCPs are fully compatible with ParaDis. For the discriminator, we instead leverage the power of parameter count and introduce parameter distribution shift.

Another objective of the ParaDis framework is space- and data-agnosticity to promote performance transfer across search spaces and tasks. While data-agnosticity can only be obtained with metrics such as `synflow`, the genetic setup ensures space-agnosticity unless additional space information is injected in the proxies.

With the knowledge that in complex conditional NAS spaces, highest parameter count does not directly translate to highest performance, we opt to define a parameter-based discriminator that defines a score around the distribution of parameter counts and targets the right model size :

$$\Phi_\alpha = e^{-(\frac{p(\alpha) - \mu}{\sigma})^2}$$

where $p(\alpha)$ is the parameter count of the considered architecture, $\mu$ and $\sigma$ are hyperparameters for calibration.

In prior work, Peng et al. (2024) used this same score to directly regularize their zero-cost proxy. Due to the complete uncertainty over what kind of distribution parameters accurately represent the space, they chose $\mu$ and $\sigma$ based on knowledge of the parameter count distribution in the search space. However, such knowledge goes against our space-agnosticity objective. Instead, we infer $\mu$ and $\sigma$ directly from the population's mean and standard deviation in the genetic algorithm at the current generation, allowing the method to remain completely agnostic to the search space. Unlike Peng et al. (2024), who directly summed their parameter count regularizer with the ZCP, we instead give it a standalone role as a discriminator metric.

At the beginning of the search, encouraging the exploration of high-parameter-count architectures promotes population diversity. Toward the end, this bias is gradually reduced to favor architectures with lower parameter counts. To achieve this, the mean of the score distribution is adjusted according to a temperature-controlled schedule. The final score is computed as follows:

$$\Phi_{\alpha,\Pi}(\tau) = e^{-\left(\frac{p(\alpha)-(\hat{\mu}(\Pi)+\tau\hat{\sigma}(\Pi))}{\hat{\sigma}(\Pi)}\right)^2}$$

where $\hat{\mu}(\Pi)$ and $\hat{\sigma}(\Pi)$ are the estimated mean and standard deviation over population $\Pi$, and the temperature $\tau$ varies between hyperparameters $\tau_{max}$ and $\tau_{min}$ during the search, both of which are search space-agnostic. The full genetic algorithm with parameter distribution shifted discriminator is detailed in appendix A.

## 5 Experiments

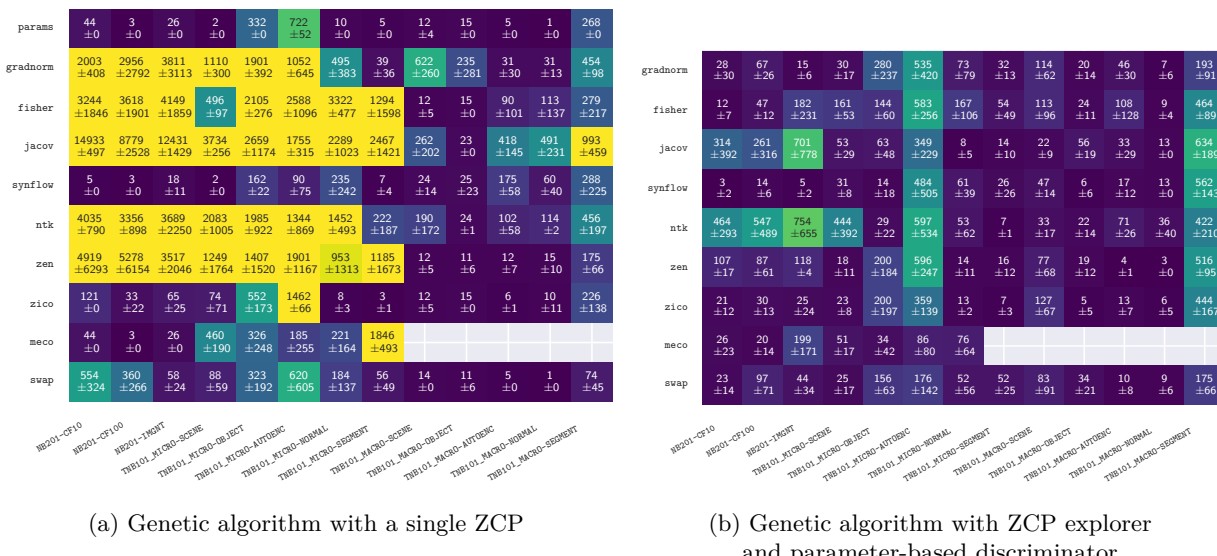

(a) Genetic algorithm with a single ZCP

(b) Genetic algorithm with ZCP explorer and parameter-based discriminator

Figure 5: Rank of best architecture in genetically evolved populations, on 8 different benchmark tasks and 10 ZCPs. We observe significant gains with a carefully designed discriminator. † The `meco` ZCP was found to be incompatible with genetic mutation in the TNB101-MACRO space.

### 5.1 Benchmark

We apply our ParaDis method to the benchmarks described in 3.1, specifically 8 tasks from the NB201 and TNB101-Micro search spaces. We use the following hyperparameters for the genetic algorithm: population size 25, sample size of 8, 15 generations with 5 evolutionary steps per generation, totaling 75 evolution steps. For ZCP computations, we use a batch size to 32. On a single NVIDIA A100 GPU, the running time of our method varies between 3 and 40 minutes depending on the task and the paired proxy. We report the ranks of the best architectures in the final populations for each search space and task in Fig. 5b. For the sake of

fair comparison, we run our genetic algorithm as a regular zero-cost method (i.e, the discriminator metric is the same as the explorer's) using the same hyperparameters and report the results in Fig. 5a.

We observe a significant boost in the quality of architectures discovered with the ParaDis discriminator compared to the single ZCPs. Specifically, the ranks of discovered architectures is lower on average for every metric. Unsurprisingly, the biggest gains are obtained for ZCPs that struggle to reach a good region of the space on their own. In accordance with our earlier benchmarking analysis, metrics which display high Spearman correlation such as `swap` do not natively translate to finding a better-ranked architecture. The results of the `synflow` metric are especially insightful as it is the only data-agnostic method studied (aside from `#params`) yet displays the most robust results natively.

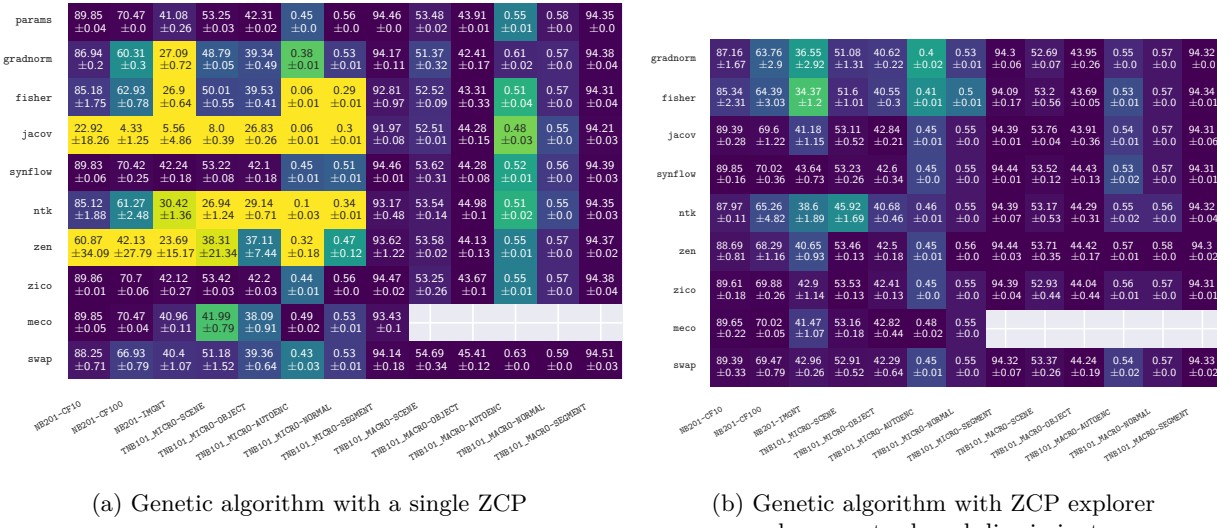

(a) Genetic algorithm with a single ZCP

(b) Genetic algorithm with ZCP explorer and parameter-based discriminator

Figure 6: Average validation accuracy in genetically evolved populations, on 8 different benchmark tasks and 10 ZCPs. † The `meco` ZCP is incompatible with genetic mutation in the TNB101-MACRO space.

We complete our benchmark analysis by reporting the average accuracy of the final population (Fig. 6). While less important than maximum accuracy, which represents the performance ceiling, high average population accuracy makes it less likely for poorly performing architectures to remain in the population after evolution. Once again, we observe significant boosts to average accuracy for metrics that displayed low average accuracy in the single ZCP setting, bringing them up to the level of better-performing metrics.

Overall, our benchmark experiments demonstrate that our parameter-based discriminator scheme restricts the search space to regions where all ZCPs can thrive, thus bridging the previously identified top rank gap. Notably, the `#params` baseline remains a dominant predictor of performance on these benchmarks. In this context, ParaDis proves highly effective at reducing the performance gap, bringing weaker ZCPs up to the level of this strong baseline. The fact that performance converges near the `#params` result suggests a saturation of these shallower search spaces, where model size is the primary driver of accuracy. This motivates the need to evaluate ParaDis in more complex environments such as the DARTS space, where structural connectivity plays a larger role than raw parameter count.

## 5.2 DARTS space experiments

In order to assess the quality of our method in a more complex setting, we conduct experiments on the standard DARTS space (Liu et al., 2019). The hyperparameters of our search are kept the same except for the number of generations which is raised to 30 for a total of 150 evolutionary steps. Generating the final population takes about 2 hours on a single NVIDIA A100 GPU. Architecture selection among the population is conducted via a simple successive halving scheme.

Table 1: Overview of CIFAR10 test performances of networks found by various NAS methods. † Indicates methods that used the DARTS search space. Training details for final architecture population in the Baseline and ParaDis methods can be found in Appendix E. All other methods list performance reported in their respective papers.

| | Validation Accuracy (%) | Time (GPU Days) | Search Type | Evaluation Type | Data-Agnostic | Space-Agnostic |
|---|---|---|---|---|---|---|
| ENAS (Pham et al., 2018) | 97.11 | 0.45 | RL | One-shot | ✗ | ✗ |
| DARTS-1st † (Liu et al., 2019) | 97.00 ± 0.14 | 0.4 | Gradient | One-shot | ✗ | ✗ |
| DARTS-2nd † (Liu et al., 2019) | 97.24 ± 0.09 | 1.0 | Gradient | One-shot | ✗ | ✗ |
| SNAS (Xie et al., 2019) | 97.15 ± 0.02 | 1.5 | Gradient | One-shot | ✗ | ✗ |
| SGAS † (Wang et al., 2021) | 97.44 ± 0.10 | 0.29 | Gradient | One-shot | ✗ | ✗ |
| DrNAS (Chen et al., 2021b) | 97.54 ±0.03 | 0.6 | Gradient | One-shot | ✗ | ✗ |
| TE-NAS † (Chen et al., 2021a) | 97.37 ± 0.064 | 0.03 | Pruning | Training-free | ✗ | ✓ |
| Zen-NAS (Lin et al., 2021) | 96.2 | - | Evolution | Training-free | ✗ | ✓ |
| MeCo † (Jiang et al., 2023) | 97.31 ± 0.05 | 0.08 | Pruning | Training-free | ✗ | ✓ |
| SWAP-NAS † (Peng et al., 2024) | 97.52 ± 0.09 | 0.004 | Evolution | Training-free | ✗ | ✗ |
| Baseline + synflow † | 97.09 ± 0.26 | 0.06 | Evolution | Training-free | ✓ | ✓ |
| Baseline + #params † | 97.06 ± 0.04 | 0.04 | Evolution | Training-free | ✓ | ✓ |
| ParaDis + synflow † | 97.29 ± 0.07 | 0.09 | Evolution | Training-free | ✓ | ✓ |

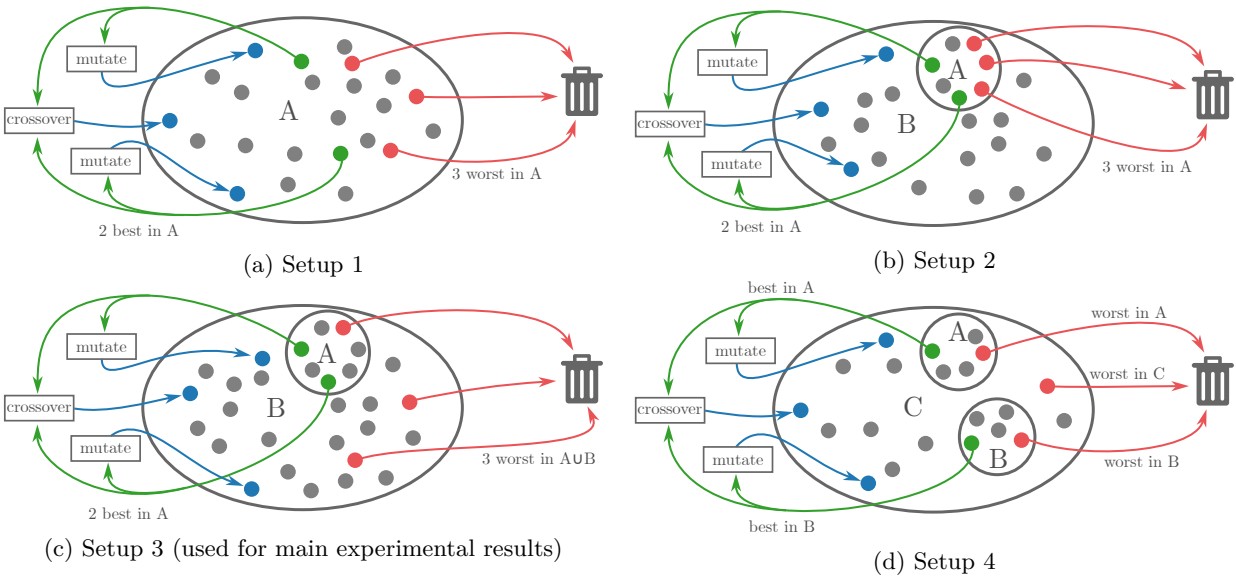

(a) Setup 1

(b) Setup 2

(c) Setup 3 (used for main experimental results)

(d) Setup 4

Figure 7: Various genetic algorithm setups compatible with ZCP-based NAS. Identifying favorable algorithmic design choices allows for fair comparison of ZCPs.

For the explorer ZCP role, we selected the synflow metric out of all available metric owing to the strong displayed results in the NAS-Bench-Suite-Zero benchmark and its data-agnostic nature which enables the construction of a fully data-agnostic and space-agnostic method inside the ParaDis framework. Our results are reported in Table 1. We also report the results of our evolutionary setup without the ParaDis discriminator paired with the synflow and #params metrics. This allows us to verify that the ParaDis coupling outperforms both of its components in the complex DARTS space. When compared with other popular and state-of-the-art NAS methods, ParaDis + synflow is competitive while remaining independant both from the data and the space.

### 5.3 Ablation on genetic algorithm design

When designing the genetic search of a zero-cost method, there are many options available. Most prior methods (Peng et al., 2024; Lin et al., 2021; Li et al., 2023) have opted for some form of tournament selection. However, the exact impact of algorithmic design choices within the frame of tournament selection has not been extensively studied.

In this section, we evaluate the performance of state-of-the-art ZCPs under various genetic setups inspired by common design choices encountered in the literature. We identify 4 different scenarii to evaluate which components of the tournament selection framework are integral to a ZCP method's performance. Graphic representations of each setup are also depicted in Fig. 7:

- Setup 1: no tournament selection. At each evolutionary step, the 2 best architectures in the whole population go on to mutation and crossover while the 3 worst architectures are discarded. Even before running any experiments, the lack of stochasticity of this scenario is questionable and could lead to early collapse in local minima.

- Setup 2: full tournament selection. At each evolutionary step, tournament selection is conducted on a subset of the population. The 2 best architectures in the subset go on to mutation and crossover while the 3 worst architectures in the subset are discarded.

- Setup 3: partial tournament selection. At each evolutionary step, tournament selection is conducted on a subset of the population. Compared to setup 2, the worst 3 architectures across the whole population are discarded, increasing the likelihood of low quality candidates being removed from the genetic pool in early stages. This is the genetic setup used throughout our main experiments.

- Setup 4: partial multi-tournament selection. At each evolutionary step, tournament selection is conducted separately on 2 subsets of the population. Winners of both subsets undergo mutation and crossover with each other. Discarded architectures are selected from both tournaments and the remainder of the population.

Table 2: Maximum accuracy in the population after ZCP evolution in NB201/ImageNet under various genetic setups.

|  | Setup A | Setup B | Setup C | Setup D |
|---|---|---|---|---|
| synflow | $44.94 \pm 0.13$ | $45.26 \pm 0.05$ | $\mathbf{46.63 \pm 0.59}$ | $46.41 \pm 0.69$ |
| swap | $44.94 \pm 0.13$ | $45.26 \pm 0.05$ | $46.14 \pm 0.46$ | $\mathbf{46.24 \pm 0.47}$ |
| meco | $44.94 \pm 0.13$ | $45.26 \pm 0.05$ | $45.20 \pm 1.45$ | $\mathbf{45.97 \pm 0.35}$ |

Table 3: Average accuracy in the population after ZCP evolution in NB201/ImageNet under various genetic setups.

|  | Setup A | Setup B | Setup C | Setup D |
|---|---|---|---|---|
| synflow | $41.03 \pm 0.34$ | $39.74 \pm 0.92$ | $\mathbf{41.54 \pm 1.42}$ | $39.70 \pm 1.71$ |
| swap | $41.03 \pm 0.34$ | $39.74 \pm 0.92$ | $42.31 \pm 0.73$ | $\mathbf{43.09 \pm 0.17}$ |
| meco | $41.03 \pm 0.34$ | $39.74 \pm 0.92$ | $\mathbf{42.48 \pm 0.81}$ | $39.83 \pm 1.44$ |

We conduct evolution for 50 evolutionary steps on the NB201/ImageNet task over 3 seeds with the `synflow`, `swap` and `meco` metrics as fitness scores and report the results for maximum and average accuracy of the population in tables 2 and 3. We observe that setups A and B produce worse results than setups C and D overall. More worryingly the evolutions collapse to the exact same populations for a given seed no matter the ZCP, indicating that the design of the genetic algorithm fully overwrites the additional information provided by the metric and leads to a local minimum, thereby creating a performance ceiling. This highlights the importance of proper genetic algorithm design as seemingly small changes (e.g where discarded architectures are selected) can be the difference maker between effective evolution and non-stochastic collapse. Furthermore, setup D does not meaningfully improve performance compared to setup C based on $1\sigma$ rules, therefore we use setup C in our experiments as it is the least complex of the two.

## 6 Conclusion

In this work, we introduce ParaDis, a framework to associate zero-cost proxies (ZCPs) with space-agnostic parameter-based constraints. Through a novel perspective on benchmarking, we identify the inability of ZCPs to correlate with the ranking of the best architectures in the space as the top rank gap and show that ParaDis improves this ongoing issue of the literature by enabling all proxies to reach suitable regions of the space. Concurrently, we show that ParaDis enables the construction of a fully data- and space- agnostic pipeline which can compete with other state-of-the-art methods on the DARTS search space. Owing to its orthogonality with ZCP design and lack of assumptions on the search space, ParaDis is a robust and versatile framework for zero-shot NAS.

**Acknowledgments**

This work was performed using HPC resources from GENCI–IDRIS (Grant 2023-AD011013972)

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

# A   Full ParaDis algorithm

---

**Algorithm 1 Para**meter **Dis**tribution Shift NAS (**ParaDis**)

---

**Require:** : population size $P$, sample size $p < P/2$, number of generations $N$, evolution steps per generation $S$, zero-cost metric $\mathcal{M}$, temperature schedule $[\tau_0, \tau_1, \ldots, \tau_N]$

1: $population \leftarrow \emptyset$
2: **while** $population.size < P$ **do**
3:     Randomly sample $arch$ from the space
4:     $arch.zcp\_score \leftarrow \mathcal{M}(arch)$
5:     $arch.params \leftarrow CountParams(arch)$
6:     Add $arch$ to $population$
7: **end while**
8: $\mu, \sigma \leftarrow Mean(population.params), STD(population.params)$
9: **for** $arch$ in $population$ **do**
10:     $arch.paradis\_score \leftarrow \Phi(arch.params, \mu, \sigma, \tau_0)$ 4
11: **end for**
12: **for** $n = 1, 2, \ldots, N$ **do**
13:     $\tau \leftarrow \tau_n$
14:     **for** $s = 1, 2, \ldots S$ **do**
15:         Discard 3 worst $archs$ from $population$ by $paradis\_score$
16:         $candidates_A \leftarrow p$ random $archs$ of $population$
17:         $candidates_B \leftarrow$ all remaining $archs$ in $population$
18:         Create children by mutation from best $archs$ in $candidates_A$ and $candidates_B$ by $zcp\_score$
19:         Create child by crossover between best $archs$ in $candidates_A$ and $candidates_B$ by $zcp\_score$
20:         Compute $zcp\_score$ and $params$ for new children
21:         Add children to $population$
22:     **end for**
23:     $\mu, \sigma \leftarrow Mean(population.params), STD(population.params)$
24:     **for** $arch$ in $population$ **do**
25:         $arch.paradis\_score \leftarrow \Phi(arch.params, \mu, \sigma, \tau)$ 4
26:     **end for**
27: **end for**

---

# B   Additional ablations

In the following sections, we extensively study the impact of the various hyperparameters on the outcome of our method. Where applicable, we conduct ablation studies using the `synflow` metric on the NB201 ImageNet16-120 task. Note that the set of hyperparameters obtained for `synflow` is not necessarily optimal for all other ZCPs. Therefore, we assume fairness in benchmarking ZCPs with the same evolutionary budget.

### B.0.1   Temperature schedule

In order to assess the sensitivity of our method to the single additional parameter of temperature scheduling, we conduct grid search over the pairs of start and end $\tau$ for the linear schedule. We report our results as heatmaps for both relevant population metrics : average and maximum population accuracy (Fig. 8). We find that the accuracy of the best evolved architecture increases with higher values of start $\tau$ and decreases with lower values of end $\tau$. Meanwhile, average population accuracy increases drastically at the mark of start $\tau = 1.5$. This ablation highlights the delicate balance at play when augmenting ZCPs with parameter count. While the best architectures have high parameter count, many high parameter count architectures display lesser performance. Therefore, tuning the schedule to favor large architectures hurts population quality. In light of these results, we adopt a linear schedule starting at $\tau = 1.5$ and ending at $\tau = 0.5$ in our experiments.

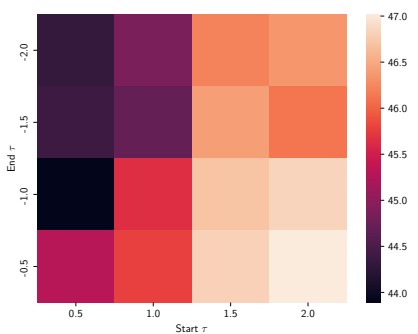

(a) Max population accuracy under various start/end $\tau$ pairs.

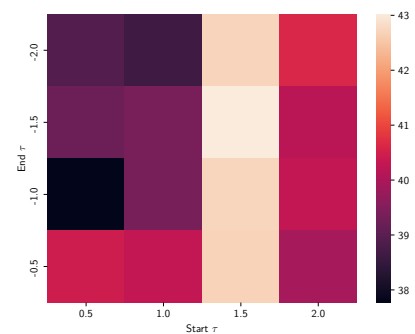

(b) Average population accuracy under various start/end $\tau$ pairs.

Figure 8: Maximum and average accuracy of populations searched using discriminator-based search with `synflow` ZCP on the NB201 ImageNet16-120 task under various temperature schedules.

### B.0.2 Number of generations

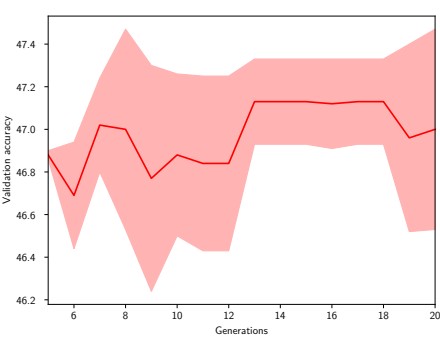

(a) Max population accuracy across various numbers of generations.

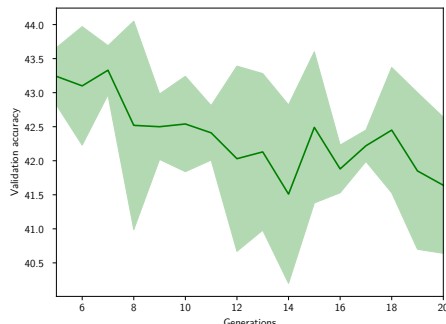

(b) Average population accuracy across various numbers of generations.

Figure 9: Maximum and average accuracy of populations for various numbers of search generations.

We conduct ablation studies over the number of generations and observe the trend of average and maximum population accuracy over the generations (Fig. 9). We observe that maximum accuracy plateaus between the 13th and 18th generation. Surprisingly, the average accuracy of the population follows a downward trend when the maximum number of generation increases. We hypothesize that while a sufficient number of iterations is required for top-ranking architectures to be evolved, longer exploration also leads to more architectures with diverse performance, reducing average population accuracy. This highlights a no free lunch tradeoff as regards the highest potential performance versus the difficulty of the search in the end population. Consequently, we adopt a generation count of 15 in our experiments which we find to yield high improvement in our experiments 5.1.

## C   Search space encodings

In this section, we detail the encodings of the search spaces used for the experiments, largely based on the implementation from Krishnakumar et al. (2022).

### C.1 NAS-Bench-201

The genome encoding of architectures from the NB201 space (Dong & Yang, 2020) is directly inferred from their graph-like cell structure. The cell is directly copied and stacked to form a complete network, therefore it is sufficient to encode a single cell. Each edge of the cell's graph is assigned to a genome position. The operation choices of the space (5 possible operations) are encoded into each position as a numerical value. With 6 edges in the graph, the resulting genome is a 6 values long string.

### C.2 TransNAS-Bench-101 Micro

Due to being based directly on the same setting, the genome encoding of architectures for the TNB101-Micro space (Duan et al., 2021) is similar to the encoding for the NB201, with 6 edges and 4 operation choices.

### C.3 TransNAS-Bench-101 Macro

The TNB101-Macro space (Duan et al., 2021) consists in a stack of 4 to 6 residual blocks where the locations of up to 4 downsamplings and up to 3 channel doublings can be chosen. It is therefore encoded as a string of 6 values, with each value assigned to one of the possible combinatorial choices: nothing, downsampling, double channels or downsampling+double channels. If the sampled architecture is smaller than the maximum size 6, the genome is padded with 0s to reach 6 values.

### C.4 DARTS

The DARTS space (Liu et al., 2019) consists in a stack of 20 cells separated into two types: normal and reduction, both of which are searchable and hold a different genome. Each 3 normal cells in the stack are followed by a reduction cell which includes a downsampling. The cells are constructed from 2 input nodes (corresponding to the outputs of cells $k - 1$ and $k - 2$, thus introducing long-range skips) and 4 intermediate nodes. The output of the cell is obtained by concatenation of the features in all intermediate nodes. Therefore, there are 14 edges per cell, however the space includes a constraint for 2 input edges per intermediate node to be non-zero, for a total of 8 non-zero edges per cell. Therefore, during sampling, mutation and crossover, we perform additional checks to ensure that the adequate number of zeroed edges is maintained. We then represent the DARTS genome as a concatenation of the normal and reduction cell genomes.

## D ZCP details

In this section, we present the definitions of each benchmarked ZCP based on theoretical insights and their implementation.

### D.1 `#params`

`#params` is defined simply as the number of trainable parameters in the network. We directly use the network implementation to count its parameters without need for additional data.

### D.2 `jacov`

`jacov` is introduced in Mellor et al. (2021) as the standalone method NASWOT then later renamed as `jacov` in Abdelfattah et al. (2021). It is defined based on the covariance matrix of the Jacobian:

$$C = cov(\nabla_w(\mathcal{L}))$$

where $\mathcal{L}$ is the loss. The Jacobian is obtained by concatenation of the gradients of the model from a backward pass over a single batch of data. Then, `jacov` is constructed as a measure of the ability of the model to

differentiate inputs in the batch:

$$\texttt{jacov} = -\sum_{i=1}^{n} \left( log(\lambda^i) + \frac{1}{\lambda^i} \right)$$

where $\lambda_1 \ldots \lambda_n$ are the eigenvalues of $C$.

### D.3 `grad_norm`

`grad_norm` is introduced by Abdelfattah et al. (2021) as a naive baseline for ZCPs. When the norm of the gradients of the model is higher, the model intuitively progresses faster in its parameter space, and is therefore easier to train.

$$\texttt{grad\_norm} = \|\nabla_w(\mathcal{L}(m, w))\|_{\mathcal{F}}$$

### D.4 `snip, grasp, synflow`

`snip` (Lee et al., 2019b) `grasp` (Wang et al., 2020) and `synflow` (Tanaka et al., 2020) are saliency metrics introduced in the pruning literature to determine the importance of specific parameters. They are aggregated over the entire network to provide a measure of the overall saliency of parameters in the model.

$$\texttt{snip} = \sum_{i}^{N} |\frac{\partial \mathcal{L}}{\partial \theta_i} \odot \theta_i|$$

$$\texttt{grasp} = \sum_{i}^{N} - \left( H \frac{\partial \mathcal{L}}{\partial \theta_i} \right) \odot \theta_i$$

$$\texttt{synflow} = \sum_{i}^{N} |\frac{\partial \mathcal{L}}{\partial \theta_i} \odot \theta_i|$$

All 3 metrics are summed over the $N$ layers of the network. $\theta_i$ are the trainable parameters at layer $i$, $\mathcal{L}$ is the loss and $\odot$ is the Hadamard product. In the case of `grasp`, the Hessian $H$ is computed via the $\mathcal{R}\{\cdot\}$ trick (Pearlmutter, 1994). `synflow` is similar to `snip`, but is data-agnostic as it does not compute over a batch of data, instead using a full matrix of 1 as input to estimate the synaptic flow.

### D.5 `fisher`

`fisher` is introduced as a ZCP by Abdelfattah et al. (2021). It is based on the aggregated sum of layerwise Fisher information (Theis et al., 2018; **?**) over the $N$ layers of the model:

$$\texttt{fisher} = \sum_{i}^{N} \left( \frac{\partial \mathcal{L}}{\partial z_i} z_i \right)^2$$

where $z_i$ is the aggregation of post-activations at layer $i$ and $\mathcal{L}$ is the loss.

### D.6 `ntk`

`ntk` is introduced as a ZCP in the TE-NAS method (Chen et al., 2021a) where it is paired with the number of linear regions. It is defined as the condition number of the Neural Tangent Kernel (NTK) (Jacot et al., 2018; Lee et al., 2019a; Xiao et al., 2019). Networks with lower condition number have better convergence property and trainability. We estimate the Neural Tangent Kernel following the TE-NAS estimation scheme:

$$\Theta = \nabla_w(\mathcal{L}^T \nabla_w(\mathcal{L})$$

where $\mathcal{L}$ is the loss and $w$ are the trainable parameters of the model. Considering the eigenvalues $\lambda_0 \ldots \lambda_n$ of $\Theta$, we have:

$$\texttt{ntk} = \lambda_n / \lambda_0$$

### D.7 `zen`

`zen` is introduced in Lin et al. (2021) as a ZCP based on the noise error of the model and the statistics of the batch normalization layers (BN). Therefore, it is specific to search spaces that contain batch normalizations.

Let $x, \epsilon$ the inputs of `zen` sampled from a normal distribution $\mathcal{N}(0, 1)$. For model $f$ and hyperparameter $\alpha$, we have:

$$\Delta = \mathbb{E}_{x,\epsilon} \|f(x) - f(x + \alpha\epsilon)\|_F$$

And for all BN layer in the model, we define:

$$\bar{\sigma} = \sum_i log \left( \sqrt{\sum_j^m \sigma_{i,j}^2 / m} \right)$$

which considers the standard deviation $\sigma_{i,j}$ of BN layer $i$ at the $j$th out of $m$ channels.

Both parts of the metric are then summed:

$$\texttt{zen} = log(\Delta) + \bar{\sigma}$$

### D.8 `zico`

`zico` (Li et al., 2023) is introduced to evaluate the generalization capacity of architectures (**??**) based on the statistics of the gradients:

$$\texttt{zico} = \sum_{i=1}^{N} log \left( \sum_{\omega \in \theta_i} \frac{\mathbb{E}\|\nabla_\omega \mathcal{L}\|}{\sqrt{Var(\nabla_\omega \mathcal{L})}} \right)$$

where $\theta_i$ are the trainable parameters at layer $i$ out of $N$ layers, and $\mathcal{L}$ is the loss.

### D.9 `meco`

`meco` (Jiang et al., 2023) is introduced based on similar insights as `zico` by noticing that the Pearson correlation matrix is closely related to the Gram matrix. Let the Pearson correlation matrix over the outputs of mode $f$ with input $X$:

$$[\mathcal{P}(X)]_{i,j} = \frac{\mathbb{E}\|(f(X_i) - \mu_{f(X_i)})(f(X_j) - \mu_{f(X_j)})\|}{\sigma_{f(X_i)} \sigma_{f(X_j)}}$$

where $\mu_x$ and $\sigma_x$ are the mean and standard deviation of $x$, respectively. `meco` is then computed based on the minimum eigenvalue $\lambda_{min}$ of the Pearson correlation matrix:

$$\texttt{meco} = \sum_{i=1}^{N} \lambda_{min}(\mathcal{P}(X))$$

### D.10 `swap`

`swap` (Peng et al., 2024) is a reframing of the concept of linear regions partition in the latent space of the network to measure the expressivity of the network (Raghu et al., 2017; Xiong et al., 2020; Chen et al., 2021a).

The set of sample-wise activation patterns of the network is gathered as:

$$\hat{\mathbb{A}}_\theta = \left\{ \mathbb{1}(p_s^{(v)})_{s=1}^S, v \in \{1, \dots, V\} \right\}$$

where $p_s^{(v)}$ is the post-activation of sample $s$ at value $v$. $\hat{\mathbb{A}}_\theta$ contains the one-hot encodings of all activation patterns obtained in the network for the samples in batch $S$.

`swap` is then equal to the number of activation patterns, which is the cardinal of the set:

$$\texttt{swap} = |\hat{\mathbb{A}}_\theta|$$

# E    Training details

In this section, we detail the training of the final population in the DARTS space on the CIFAR10 dataset to obtain our experimental results reported in Table 1.

The training protocol follows the standard DARTS training scheme as intrdoduced by Liu et al. (2019). Specifically, the generated candidate networks are trained for 600 epochs using SGD with the following hyperparameters: batch size 96, model width 36. The optimizer is SGD with momentum 0.9 and weight decay $3 \times 10^{-4}$. The initial learning rate of 0.025 decreases to 0 following a cosine annealing schedule. Additional training operations include cutout (DeVries & Taylor, 2017), path dropout with probability 0.2 and auxiliary head with weight 0.4.

# F    Benchmarks for additional top cutoffs

In Figures 10 and 11 and we report additional benchmarks restricted to the top 2% and top 5% of the search spaces. The results demonstrate that the correlation collapse linked with the top-rank gap do not happen after at a single threshold, instead displaying a gradual collapse as the quality of architectures increases.

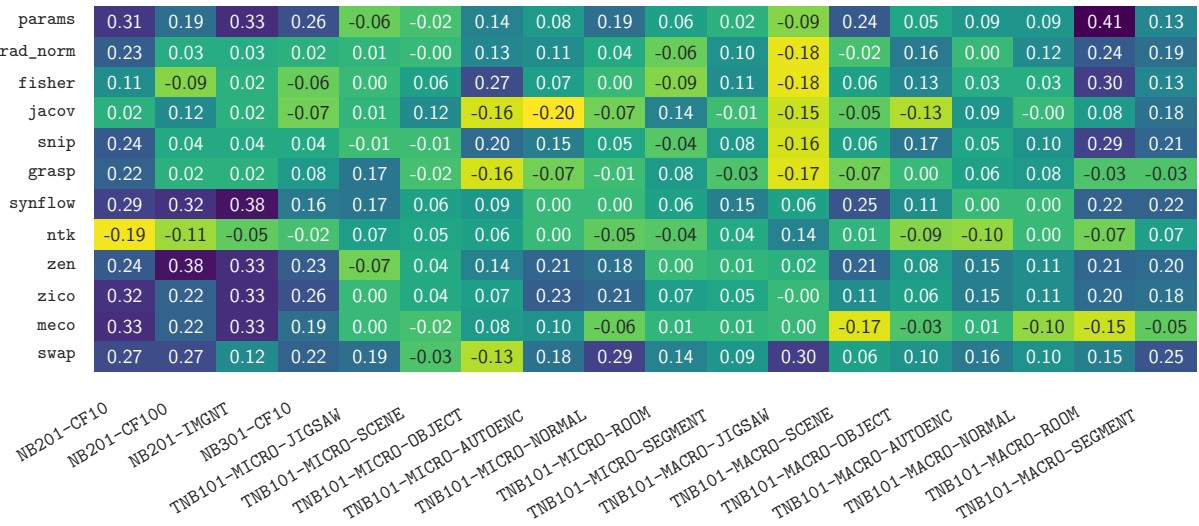

Figure 10: Kendall-$\tau$ correlations of various ZCPs on the NB201, NB301 and TNB101 benchmarks. Kendall-$\tau$ was computed on 3 seeds against the top 2% architectures in the space.

| | NB201-CF10 | NB201-CF100 | NB201-IMGNT | NB301-CF10 | TNB101-MICRO-JIGSAW | TNB101-MICRO-SCENE | TNB101-MICRO-OBJECT | TNB101-MICRO-AUTOENC | TNB101-MICRO-NORMAL | TNB101-MICRO-ROOM | TNB101-MICRO-SEGMENT | TNB101-MACRO-JIGSAW | TNB101-MACRO-SCENE | TNB101-MACRO-OBJECT | TNB101-MACRO-AUTOENC | TNB101-MACRO-NORMAL | TNB101-MACRO-ROOM | TNB101-MACRO-SEGMENT |
|---|---|---|---|---|---|---|---|---|---|---|---|---|---|---|---|---|---|---|
| params | 0.10 | 0.26 | 0.38 | 0.27 | -0.07 | -0.12 | 0.16 | -0.04 | 0.04 | 0.23 | 0.09 | 0.22 | 0.22 | 0.23 | 0.19 | 0.23 | 0.20 | 0.10 |
| grad_norm | -0.15 | -0.05 | 0.17 | -0.03 | 0.02 | 0.08 | 0.18 | -0.07 | 0.03 | 0.29 | 0.22 | -0.08 | 0.11 | 0.15 | 0.07 | 0.17 | 0.20 | -0.01 |
| fisher | -0.24 | -0.13 | 0.12 | -0.11 | 0.00 | 0.10 | 0.27 | -0.05 | 0.20 | 0.28 | 0.11 | -0.02 | 0.16 | 0.17 | 0.10 | 0.12 | 0.23 | -0.03 |
| jacov | 0.12 | 0.03 | 0.06 | -0.04 | 0.12 | 0.10 | -0.20 | -0.01 | 0.01 | 0.09 | 0.01 | -0.11 | -0.01 | -0.01 | 0.16 | 0.12 | -0.03 | 0.05 |
| snip | -0.14 | -0.05 | 0.17 | -0.02 | 0.01 | 0.07 | 0.21 | -0.08 | 0.04 | 0.28 | 0.22 | 0.01 | 0.17 | 0.20 | 0.13 | 0.23 | 0.24 | 0.06 |
| grasp | -0.14 | -0.08 | 0.10 | 0.13 | 0.02 | -0.09 | -0.16 | 0.01 | -0.03 | -0.16 | -0.00 | -0.07 | -0.04 | 0.00 | 0.09 | 0.01 | -0.08 | 0.06 |
| synflow | 0.22 | 0.40 | 0.46 | 0.10 | 0.02 | 0.11 | 0.21 | 0.00 | 0.00 | 0.30 | 0.31 | 0.15 | 0.25 | 0.19 | 0.00 | 0.00 | 0.11 | 0.23 |
| ntk | -0.14 | -0.10 | -0.08 | 0.14 | -0.00 | 0.05 | -0.00 | 0.04 | 0.02 | -0.02 | 0.00 | 0.16 | 0.06 | -0.01 | -0.07 | 0.04 | -0.08 | -0.12 |
| zen | 0.28 | 0.45 | 0.32 | 0.26 | -0.06 | 0.04 | 0.22 | -0.04 | 0.03 | 0.20 | 0.12 | 0.14 | 0.21 | 0.20 | 0.23 | 0.24 | 0.11 | 0.23 |
| zico | 0.06 | 0.22 | 0.41 | 0.27 | -0.05 | 0.10 | 0.21 | -0.06 | 0.04 | 0.18 | 0.16 | 0.08 | 0.15 | 0.17 | 0.23 | 0.22 | 0.11 | 0.23 |
| meco | 0.14 | 0.28 | 0.38 | 0.20 | 0.00 | 0.02 | 0.23 | -0.06 | -0.03 | 0.06 | -0.01 | 0.00 | -0.13 | -0.01 | 0.01 | 0.07 | -0.18 | -0.06 |
| swap | 0.06 | 0.23 | 0.31 | 0.27 | 0.09 | 0.03 | 0.00 | -0.11 | 0.02 | 0.20 | 0.19 | 0.49 | 0.09 | 0.32 | 0.38 | 0.36 | 0.00 | 0.30 |

Figure 11: Kendall-$\tau$ correlations of various ZCPs on the NB201, NB301 and TNB101 benchmarks. Kendall-$\tau$ was computed on 3 seeds against the top 5% architectures in the space.

# G   Kendall-$\tau$ benchmark

While Spearman rank correlation is the most commonly used benchmarking metric for ZCPs, it only captures linear codependancies directly on accuracy values. For the NAS purpose of finding the best architecture overall in the search space, ZCPs can also be evaluated based on their raw ranking ability, which can be done via Kendall-$\tau$:

$$\tau = \frac{2}{n(n-1)} \sum_{i<j} \text{sign}(x_i - x_j)\, \text{sign}(y_i - y_j)$$

where $n$ is the size of the search space.

In Figures 12 and 13 we report benchmarks for the full search space and top 1% of the space similar to Figures 2 and 3. We observe a similar collapse as with the Spearman benchmark, providing additional evidence of the top-rank gap.

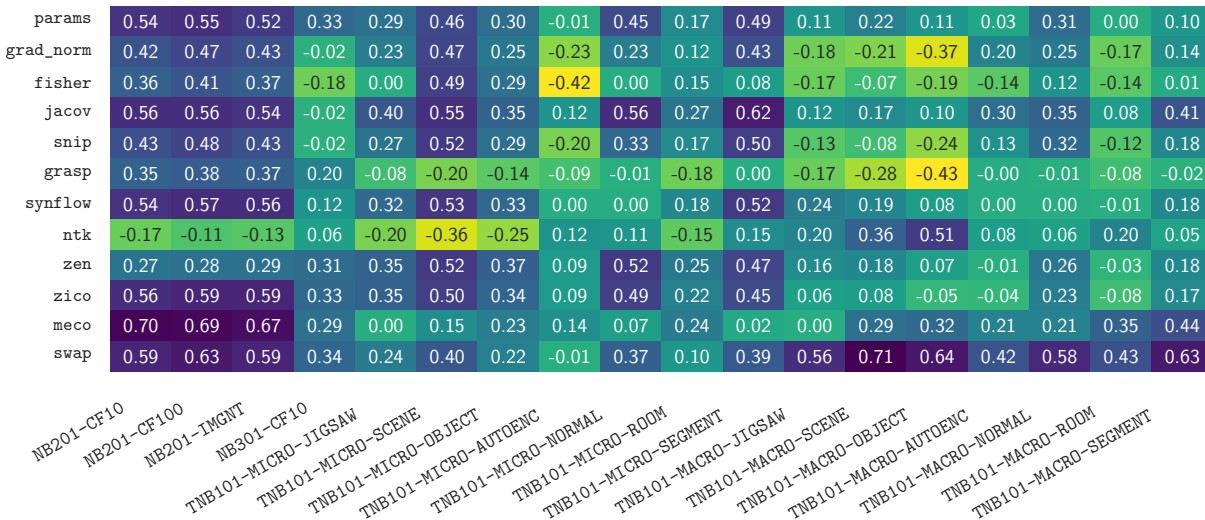

Figure 12: Kendall-$\tau$ correlations of various ZCPs on the NAS-Bench-201, NAS-Bench-301 and TransNAS-Bench-101 benchmarks. Kendall-$\tau$ was computed over 3 seeds against the entire search space.

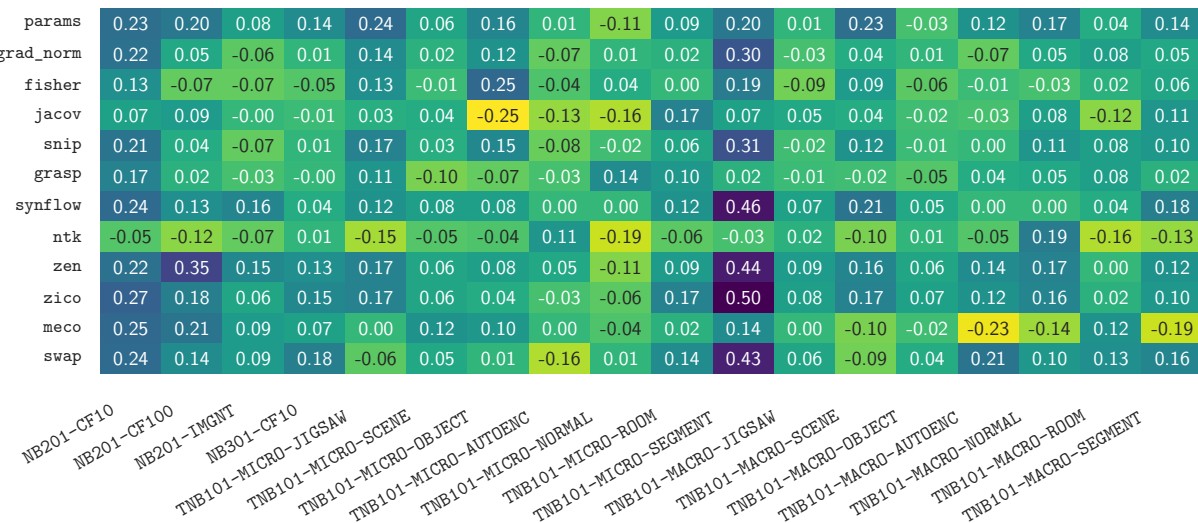

Figure 13: Kendall-$\tau$ correlations of various ZCPs on the NB201, NB301 and TNB101 benchmarks. Kendall-$\tau$ was computed on 3 seeds against the top 1% architectures in the space.

## H    More details about the Spearman correlation benchmarks

In the main paper, confidence intervals for the Spearman correlation benchmarks are omitted to keep the table readable. The displayed value is averaged over 3 seeds. In the following tables we present the Spearman correlation for both the full space setting and the top 1% with $1\sigma$ error.

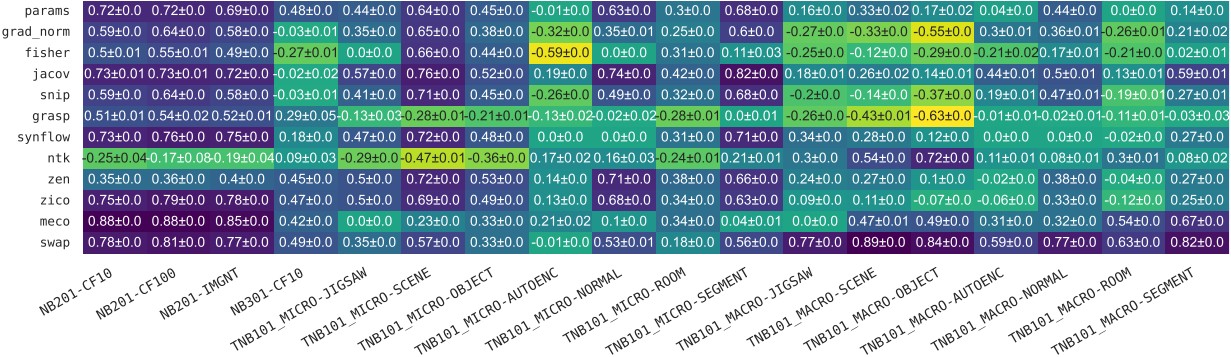

Figure 14: Spearman rank correlations of various ZCPs on the NAS-Bench-201, NAS-Bench-301 and TransNAS-Bench-101 benchmarks. Spearman was computed over 3 seeds against the entire search space.

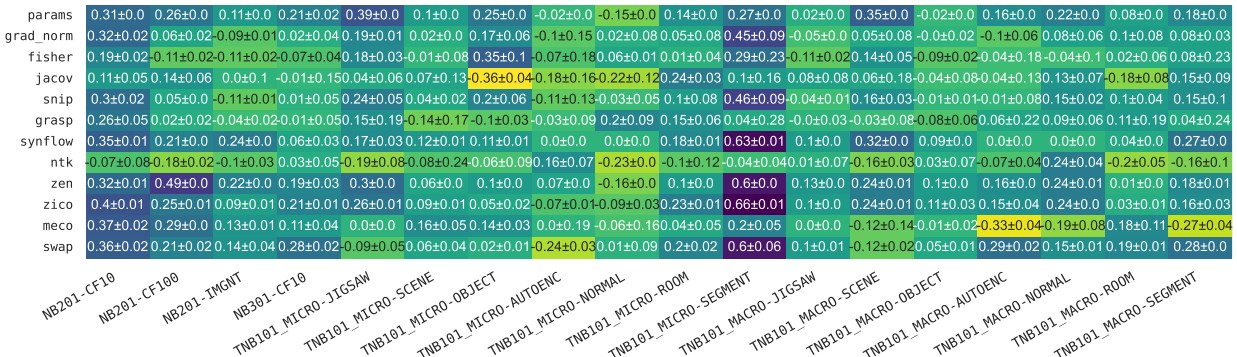

Figure 15: Spearman rank correlations of various ZCPs on the NB201, NB301 and TNB101 benchmarks. Spearman was computed on 3 seeds against the top 1% architectures in the space.

