# OpenReview forum: "Insights From a Data- and Space-Agnostic Approach to Zero-Cost Proxies"
_TMLR — Accepted by TMLR_

### Review · Reviewer_ox9M · 2026-02-22

**Summary Of Contributions:**

The paper studies the limitations of zero-cost proxies (ZCPs) for training-free neural architecture search and identifies a key issue: although ZCPs correlate well with performance across entire search spaces, they fail to reliably rank the top architectures (the “top-rank gap”). To address this, the paper proposes ParaDis, a NAS framework that assigns ZCPs an exploration role while using an adaptive, parameter-count-based discriminator to constrain the search space in a space-agnostic way. Experiments on NAS-Bench-Suite-Zero and DARTS show that ParaDis improves the quality of architectures found by various ZCPs and enables competitive fully data- and space-agnostic search.

**Audience:**

Yes

**Audience Explanation:**

The paper addresses zero-cost proxies and training-free neural architecture search. These topics are directly relevant to parts of the TMLR audience working on NAS, model selection, efficiency, and meta-learning. In particular, the identification of the top-rank gap and the critique of standard correlation-based benchmarking provide useful empirical insight for researchers developing or evaluating new ZCPs.
While the proposed method (ParaDis) is relatively simple and incremental, the diagnostic analysis and benchmarking perspective would likely be of interest to researchers focused on robust evaluation methodologies and lightweight NAS.

**Broader Impact Concerns:**

I do not identify any specific ethical or societal risks associated with this work. The paper proposes a methodological improvement for training-free neural architecture search and evaluates it on standard benchmarks . The work is purely algorithmic and does not involve sensitive data, human subjects, or direct deployment considerations. I therefore consider broader impact concerns to be minimal and no additional Broader Impact Statement is required.

**Claims And Evidence:**

Yes

**Claims Explanation:**

The main empirical claim that ZCPs have strong global correlation but degrade significantly on top-performing architectures (“top-rank gap”) is clearly supported by the reported full-space vs. top-1% Spearman results across benchmarks . The claim that ParaDis improves the quality of architectures found by ZCPs is also supported by best-rank and mean population accuracy results, along with ablations on GA design and temperature scheduling . However, the results indicate that parameter count is a dominant factor in several spaces; ParaDis often brings weaker ZCPs closer to the strong #params baseline . The DARTS gains are competitive but modest . Overall, the core claims are supported, though some conclusions could be framed more cautiously.

**Requested Changes:**

1. Please provide the exact training/evaluation recipe used for the final selected architectures (epochs, optimizer, augmentation, split, whether “validation accuracy” is the reported metric), and explicitly state whether the numbers in Table 1 are directly comparable to prior work or no.

2. Please inlcude the code in the submission. It helps reproducability and it is a standard practice in the ML community.

3. In addition to top-1%, could you please report at least one or two other cutoffs (e.g., top-0.5%, 2%, 5%) to show the phenomenon is not an artifact of a single threshold.

4. the abstract mentions that the method is "competitive against ...". It is a common practice to report some actual numbers (for example the method is x % better than SOTA etc). Please edit the abstract accordingly.

5. The related work ctin includes works since 2021, 2023 and the most recent is 2024. Given the fact that it is a rapidly evaloving field, could you please check if there is any more recent work (for example 2025) and add/compare it ?

6. In Figure 5 and 6, would it be possible to add results for the discriminator-only baseline (#params-only GA) within the same setup to further contextualize ParaDis gains across all tasks?

7. How does ParaDis compare against ZCP ensembling strategies (e.g., Bayesian-optimized linear combos or random-forest-based predictors) when used as the explorer within the same GA?

---

> ### Author Response · Authors · 2026-03-27
>
> We thank the reviewer for the thorough and constructive feedback. As for the requested changes, we address them below.
>
> 1 - We have included the training parameters of the DARTS evaluation protocol in appendix E. For the sake of clarity, we indicate that the numbers reported in Table 1 for ParaDis as well as the two baselines follow the standard DARTS evaluation protocol using the original DARTS codebase and hyperparameters: the selected architecture is trained from scratch for 600 epochs with batch size 96, width 36, using SGD with momentum 0.9, cosine annealing, cutout regularization and auxiliary towers. The reported metric is test accuracy. For all other methods displayed in Table 1, we reported numbers taken directly from their respective original papers, most of which were under a similar evaluation protocol. We will also add this clarification to the Table 1 caption.
>
> 2 - On the topic of code availability, we are currently cleaning up the codebase and will be adding it as a supplementary zip archive by next week, then release it as a public github repository. Our code is directly forked from NASLib (https://github.com/automl/NASLib/tree/zerocost) and covers all newer ZCP implementations, the ParaDis genetic algorithm and additional mutation and crossover logic for the TNB101 space.
>
> 3 - Thank you for this suggestion. We will be running the top-0.5%, top-2% and top-5% benchmarks will report the results in a followup post, then include them in the appendix.
>
> 4 - We will be modifying the abstract to state that our method achieves a CIFAR-10 test accuracy of 97.29 ± 0.07% in the DARTS space, within .25% of the state-of-the-art, remaining competitive despite the space-agnostic and data-agnostic constraints.
>
> 5 - Owing to the rapid evolution of the field, we are conducting additional survey of recent 2025 papers and preprints and updating the related work section and citations accordingly. This includes all works suggested by reviewer WaJS, for which we will thoroughly evaluate whether they should be mentioned in the related works section. We will survey additional recent works that may have escaped our radar. To the best of our knowledge, no specific ZCP has reached state-of-the-art compared to the proxies we employed. However, since our method is largely orthogonal to ZCP design, we are paying specific attention to papers in the general zero-cost NAS sphere, including methods for ZCP ensembling which have seen some advances as pointed out by other reviewers.
>
> 6 - Could you clarify that you are suggesting to run #params as an exploration metric paired with the ParaDis discriminator? The #params-only GA can already be found on the top row in figures 5a and 6a, which suggests the efficacy of parameter-driven GA alone. This is redundant with using parameter count for exploration and ParaDis for discrimination, hence why we left it out of our experiments. Nonetheless, if you believe that this deserves to appear in the paper, we will run the associated experiment.
>
> 7 - Direct ensembling of proxies has been explored in other papers [1, 2, 3]. Some of which we cited in our paper while others were kindly brought up by other reviewers:
> - He et al. [1] proposed to fit weighted sums of ZCPs
> - Kadlecová et al. [2] used ZCPs and graph properties as input features for random forest models
> - Most similarly to our work, Lee & Ham [3] ensembled ZCPs in a genetic framework through non-linear aggregation
>
> These directions of research are quite dissimilar, or even complementary to our approach which divides the roles of ZCPs between exploration and discrimination. Not only can some of these works (e.g AZ-NAS [3]) be directly integrated in our framework, we in fact note that the question of finding optimal ensembles of metrics for both the explorer and discriminator role constitutes a significant direction of research that warrants more than a simple ablation here. Hence, we considered this to be outside the scope of our paper.
>
> References:
>
> [1] He et al., Robustifying and Boosting Training-Free Neural Architecture Search, ICLR 2024
>
> [2] Kadlecová et al., Surprisingly Strong Performance Prediction with Neural Graph Features, 2024
>
> [3] Lee & Ham, AZ-NAS: assembling zero-cost proxies for network architecture search, CVPR 2024

---

> ### Author Response · Authors · 2026-03-31
> **Paper revision**
>
> As part of our efforts to improve the paper and bring its contents up-to-date with the current state-of-the-art, we went through all of the papers suggested by reviewer WaJS.
>
> We also conducted our own survey of recent papers in the field, but did not find any of better relevancy than the ones suggested by the reviewer.
>
> We believe our related works section has been massively overhauled as a result.
>
> Additionally, we have added the benchmarks for the top 2% and top 5% cutoffs in appendix. We left out the proposed top 0.5% due to the fact that in some search spaces, the number of architectures in the space would be too little.
>
> Finally, we have added the source code for our method as supplementary material.
>
> We once again thank the reviewer for their helpful feedback and suggestions.

---

> > ### Comment · Reviewer_ox9M · 2026-04-04
> > **Minor follow-up questions**
> >
> > I appreciate the effort to address the comments, and especially the addition of training details, code, and extended top-k analyses. I have a couple of minor follow-up clarifications:
> >
> > 1. Regarding the role of parameter count: while the #params-only GA baseline is informative, my original question was more about disentangling the interaction between exploration and discrimination. In particular, it would be helpful to clarify whether using #params explicitly as the explorer together with the ParaDis discriminator leads to similar behavior, or whether ParaDis provides additional benefit beyond parameter-driven search.
> >
> > 2. On ZCP ensembling: I understand this may be outside the main scope, but a brief discussion (even qualitative) of how ParaDis relates to or could complement such approaches would strengthen the positioning of the method.

---

> ### Author Response · Authors · 2026-04-10
> **Response to follow-up questions**
>
> Thank you for the response and clarifications.
>
> 1 - We went ahead and ran experiments for the #params + ParaDis setup and got the following results (columns are in the same order as the figures in the paper):
>
> #params max accuracy
>
> | 42 | 7 | 18 | 101 | 52 | 360 | 14 | 7 | 12 | 16 | 5 | 9 | 359 |
>
> #params avg accuracy
>
> | 89.41 | 69.56 | 42.26 | 53.06 | 42.46 | 0.45 | 0.56 | 94.41 | 54.05 | 44.45 | 0.56 | 0.58 | 94.33 |
>
> So the performance of #params with ParaDis is about the same as the baseline barring minor improvements on specific tasks, but it's still inferior to synflow. Actually, the profile of the two metrics on the benchmark is quite similar, which could be explained by their data-agnostic nature.
>
> One thing to consider is how simplistic the benchmark spaces are. This further highlights the importance of testing methods in more challenging settings such as DARTS, although we unfortunately do not have the compute to run #params + ParaDis on DARTS right now.
>
> Overall, this opens up an important next direction of research which is that of adapted exploration proxy: given a setup such as ParaDis with different roles assigned to different proxies, what kind of proxy design for the explorer yields the best results (but not necessarily the best results in the baseline). This could even be a compound proxy, which ties back in to the other question.
>
> 2 - Since other reviewers brought up the topic of ZCP ensembling, we mentioned it more explicitly in the revised paper and especially reworked the related works section to properly contrast our approach against existing ensembling methods. This should make our positioning better.
>
> For for a more detailed explanation: ParaDis as a framework (not necessarily the discriminator itself) can be complementary to all other ZCP ensembling, given that they are ported over to a genetic framework. In the most direct example of AZ-NAS [1], one can simply reuse the AZ-NAS proxies and aggregation protocol and attach it to ParaDis as the explorer part, so the two methods could be directly combined with little reframing.
>
> Outside of aggregation, another idea is to conduct selection separately for all proxies and ensemble the result by majority vote. For integration with ZCPs-as-features ideas [2,3], the explorer could be derived from the output of the trained performance predictor. These connections are all possible by lightly tweaking the ParaDis framework and most importantly by keeping the explorer/discriminator divide intact, hence the complementarity of the approach.
>
> We hope these new insights respond to your interrogations.
>
> References:
> [1] Lee & Ham, AZ-NAS: assembling zero-cost proxies for network architecture search, CVPR 2024
> [2] Kadlecová et al., Surprisingly Strong Performance Prediction with Neural Graph Features, 2024
> [3] Akhauri and Abdelfattah, Encodings for prediction-based neural architecture search, ICML 2024

---

### Review · Reviewer_WaJS · 2026-03-04

**Summary Of Contributions:**

This paper provides an analysis of ZCPs and their ability to rank the top 1% in a NAS search space properly. The analysis on four search spaces (NAS-Bench-201, NAS-Bench301, TransNASBench-101-Micro, TransNASBench-101-Macro) identifies a top-rank gap, i.e., the issue of not being able to properly distinguish the top 1 architectures from each other. In order to improve on that for an actual search, this paper includes two different ZCPs into an evolutionary search algorithm, named ParaDis, which uses one ZCP as an explorer metric (a metric which is able to distinguish good architectures to decide which architectures using for mutation and crossover, here using synflow) and a discriminator metric, which decides which architectures to remove from the population, here #params). This paper applies the proposed search on NAS-Bench-201, TransNASBench-101-Micro, and TransNASBench-101-Macro as well as compares it to other NAS methods on DARTS. The proposed method shows results on par with other ZCP-based search approaches.

**Audience:**

Yes

**Audience Explanation:**

The genome description is not clear. Krishnakumar et al. (2022) reference to White et al. (2021) (see missing reference list). White et al. (2021) use the adjacency matrix of the network cells and one-hot list of operations.  In case this paper also depends on the adjacency matrix and pre-defined one-hot list of operations, this reduces the importance of this search approach, to cell-based search spaces or small macro search spaces. But hinders the possibiity to go to larger search spaces, such as einspace (Ericcson et al. (2024)) or even generalize to others.

Though the propsed search is interesting, the idea of using several ZCPs jointly and also in a search process is not new (Akhauri and Abdelfattah (2024), Kadlevocá et al. (2024), Lukasik et al. (2024), Qin et al. (2025)).  Therefore the novelty is not clear.

**Broader Impact Concerns:**

No broader impact concerns

**Claims And Evidence:**

No

**Claims Explanation:**

While this paper provides an interesting analysis about the top-rank gap, this issue is identified using the spearman-rank correlation. Providing the kendall-tau values, would be a stronger evidence and better fit here, since the latter is a metric directly focussing on the ability to rank inputs correctly. Therefore the actual outcome of the top-rank gap feels not fully correclty supported.

The statement on pages 1/2 about the correlation of ZCPs with underlying network properties was already discussed in Kadlecová et al. (2024) and shown in their respective Figure 1. With the outcome, that most ZCPs cannot distinguish structurally similar networks. Therefore the actual statemtent in this paper is not clear to me.

The statement on page 5, that NAS-Bench-101 not providing significalty more insights, should be also taken with caution, since this search space shows different ZCPs behaviour in Krishnakumar et al. (2022).

More description about why synflow was used for the ParaDis would be helfpul, since according to Figure 3, zen would be the best fit for the explorer. It has the highest absolute correlation for all considered tasks and benchmarks (it is also not data-dependent). A purely experimental reason is not sufficient enough.

**Requested Changes:**

- This paper needs to update the mentioned points under the "evidence claims."
- In addition, a differentiation from current literature and making the actual novelty clear is crucial.

- This paper needs to update the references. 20  references are wrongly cited.

- Table 1 is misleading. Since the proposed search method is bold. However this is not the best result.

- In addition there are several papers missing. Some examples:

Y. Akhauri and M. S. Abdelfattah, “Encodings for prediction-based neural architecture search,” in ICML, 2024.

M. Chen, H. Peng, J. Fu, and H. Ling, “Autoformer: Searching transformers for visual recognition,” ICCV, 2021.

P. Dong, L. Li, Z. Tang, X. Liu, Z. Wei, Q. Wang, and X. Chu, “Parzc: Parametric zero-cost proxies for efficient nas,” AAAI, 2025

H. Ji, Y. Feng, J. Fan, and Y. Sun, “Carl: Causality-guided architecture representation learning for an interpretable performance predictor,”  in ICCV, 2025

Z. Ji, G. Zhu, C. Yuan, and Y. Huang, “RZ-NAS: Enhancing LLM-guided neural architecture search via reflective zero-cost strategy,” in ICML, 2025

J. Lee and B. Ham, “AZ-NAS: assembling zero-cost proxies for network architecture search,” in CVPR, 2024.

M. Lin and J. Luo, “Per-architecture training-free metric optimization for neural architecture search,” in NeurIPS, 2025

J.Lukasik, M. Möller, and M. Keuper, “An evaluation of zero-cost proxies - from neural architecture performance prediction to model robustness,” in International Journal of Computer Vision (IJCV) 133(5): 2635-2652, 2024.

S. Qin, G. Kadlecová, M. Pilát, S. B. Cohen, R. Neruda, E. J. Crowley, J. Lukasik, and L. Ericsson.“Transferrable surrogates in expressive neural architecture search spaces,” in AutoML, 2025.

O. Tybl and L. Neumann, “Training-free neural architecture search through variance of knowledge of deep network
weights,” in CVPR, 2025.

C. White, A. Zela, R. Ru, Y. Liu, and F. Hutter, “How powerful are performance predictors in neural architecture search?” in NeurIPS, 2021.

---

> ### Author Response · Authors · 2026-03-27
>
> We thank the reviewer for the detailed feedback. We would like to address each concern below.
>
> 1 - On the topic of the chosen correlation metric. Our choice of correlation metric naturally went to Spearman rank correlation due to its prevalence within the literature, which makes our analysis more directly comparable with other popular works. We actually agree that Kendall-tau is a natural evaluation framework for ZCP benchmarking due to its higher focus on rank relationships. Which correlation metric to use exactly is a crucial topic for the ZCP subfield, but seems rather unexplored [1].
>
> We recomputed all benchmarks using Kendall-tau correlation on both the full space and top-1% settings and included the results in appendix F.
>
> The results confirm our conclusions: all ZCPs exhibit a sharp drop in correlation when restricted to the top-1% architectures. The relative ordering of the proxies is also consistent with the Spearman results.
>
> 2 - On the topic of the novelty of our work. While the underlying issues that we identify are similar to those of Kadlecová et al. [2], the exact protocol used to reveal them is different. Kadlecová et al. approached the problem from the angle of the reliance on convolution layers, while we use the benchmarking angle. Therefore, the two observations are complementary rather than redundant.
>
> Moreover, the core methodological value of ParaDis differs from the work of Kadlecová et al. [2], Qin et al. [3], Lukasik et al. [4], Akhauri & Abdelfattah [5] and Lee & Ham [6]:
> - Kadlecová et al. [2], Qin et al. [3] and Lukasik et al. [4] and Lee & Ham [6] ensembled ZCPs and graph features for exploration purposes. However, we introduced the explorer and discriminator setup which can work seamlessly with ZCP ensembling. In fact, finding optimal ensembles of metrics for our two distinct explorer and discriminator role is an interesting prospect for future work. We also argue that AZ-NAS could directly be associated with ParaDis in an orthogonal way.
> - Kadlecová et al. [2], Qin et al. [3], Lukasik et al. [4] and Akhauri & Abdelfattah [5] all worked with different search paradigms than genetic algorithms, those being random forests, accuracy prediction heads and LLM-driven surrogates.
>
> In comparison, our work introduces two novel aspects:
> - the distinction between exploration and discrimination roles in genetic algorithm frameworks.
> - the inclusion of space-agnosticity and data-agnosticity constraints in algorithm design, which tend to be overlooked yet are essential to ensure the methods' efficacy generalizes across search spaces and tasks.
>
> We acknowledge that these works are closely related to the subject at hand and will be including them in our related works section.
>
> 3 - Regarding the choice of the synflow metric. The reviewer correctly noted that zen achieves higher absolute top-1% Spearman correlation on several tasks as reported in Figure 3. Besides our preliminary empirical results, zen assumes the presence of batch normalization layers in the network, and is thus sensitive to search space changes, preventing us from realizing the fully space- and data-agnostic pipeline. It so happens that all benchmark search spaces used here contain built-in batch normalizations. However, zen cannot evaluate all architectures in spaces that lack this assumption (i.e einspace [7]).
>
> Regardless, we will be re-running ParaDis in the ParaDis + zen setting and report the results in a followup post.
>
> 4 - Regarding genome encoding. As the reviewer noted, we use the standard genome representations from Krishnakumar et al. [8]. This is a purely implementational issue. We acknowledge that this limits the direct applicability of our codebase to other, non cell-based search spaces. However, the ParaDis framework itself, including the explorer/discriminator role assignment and the adaptive parameter distribution estimator, can still be re-implemented for other paradigms.
>
> That is why we especially insist on the point of space-agnosticity (similar to encoding-agnosticity): space-agnostic methods do not assume the structure of the search space, therefore porting them from one search space to another is only a matter of implementation.

---

> > ### Author Response · Authors · 2026-03-27
> >
> > 5 - We softened our statement about NAS-Bench-101 since it was mainly motivated by computational budget constraints given the size of the space.
> >
> > 6 - We removed the bolding from Table 1.
> >
> > 7 - We will correct all citation errors in our next revision. Regarding the other citations you proposed, the ones discussed in point 2 ([3], [4], [5], [6]) should definitely be included. We will be discussing whether to add the remaining citations in a followup post. Additionally, since we are conducting a complementary survey as requested by reviewer ox9M, we will mention any other relevant paper we come across. We thank you once more for suggesting these additions.
> >
> > References:
> >
> > [1] Ly-Manson et al., Zero-Cost Benchmarks: Towards Lower Reliance on Spearman Rank Correlation, AutoML 2025 Non-Archival Content Track
> >
> > [2] Kadlecová et al., Surprisingly Strong Performance Prediction with Neural Graph Features, 2024
> >
> > [3] Qin et al., Transferrable Surrogates in Expressive Neural Architecture Search Spaces, AutoML 2025
> >
> > [4] Lukasik et al., An evaluation of zero-cost proxies - from neural architecture performance prediction to model robustness, IJCV 133(5): 2635-2652, 2024
> >
> > [5] Akhauri & Abdelfattah, Encodings for prediction-based neural architecture search, ICML 2024
> >
> > [6] Lee & Ham, AZ-NAS: assembling zero-cost proxies for network architecture search, CVPR 2024
> >
> > [7] Ericsson et al., einspace: Searching for Neural Architectures from Fundamental Operations, NeurIPS 2024
> >
> > [8] Krishnakumar et al., NAS-Bench-Suite-Zero: Accelerating Research on Zero Cost Proxies, NeurIPS Datasets and Benchmarks Track 2022

---

> > ### Author Response · Authors · 2026-03-31
> > **Overhaul of the Related Works section**
> >
> > In order to improve the paper and bring its contents up-to-date with the current state-of-the-art, we went through all of the papers suggested by the reviewer. We thank the reviewer once again for their help as the papers were for the most part very relevant to our work. This has allowed us to massively overhaul our related works section. Below we review each of the papers and whether we have added them as citations:
> >
> > - Akhauri and Abdelfattah, Encodings for prediction-based neural architecture search, ICML 2024
> >
> > In this work, the authors introduce accuracy prediction using inputs from multiple sources related to neural network architectures, including structural properties, ZCPs and encodings produced from unsupervised methods (i.e arch2vec) and supervised methods (i.e running architectures through GNNs). While the predictor paradigm differs greatly from our framework, this is an example of a recent ZCP ensembling method which has a place in our related works section.
> >
> > - Chen et al., AutoFormer: Searching Transformers for Visual Recognition, ICCV 2021
> >
> > While AutoFormer is a foundational paper for the description and search of transformer search spaces, we do not cover the case of transformers in our work hence we did not add it to our related works.
> >
> > - Dong et al., ParZC: Parametric zero-cost proxies for efficient nas, AAAI 2025
> >
> > The authors challenge the assumption made by many ZCPs that all nodes in the architecture graph are of similar importance. They consequently construct ParZC, a mixer + Bayesian network architecture directly optimized on the Kendall-tau score, which processes ZCPs at each node into a single score. While this does not seem to be the main intent of the method, this framework does ensemble ZCPs in a more roundabout way. The execution is actually quite similar to Akhauri & Abdelfattah. Therefore, we included it in our related works.
> >
> > - Ji et al., CARL: Causality-guided architecture representation learning for an interpretable performance predictor, ICCV 2025
> >
> > This work introduces a new GNN-based performance predictor pipeline which categorizes graph sub-structures into the critical and redundant types. While these insights could be reused in a ZCP context, this paper leans more towards the predictor-based side which is quite distant from our work, therefore we did not include it.
> >
> > - Ji et al., RZ-NAS: Enhancing LLM-guided neural architecture search via reflective zero-cost strategy, ICML 2025
> >
> > In this work, the authors use ZCPs as context to instruct LLM-driven iterative search. Since this is a novel and effective paradigm for zero-cost methods, we included it in the zero-cost section of our related works.
> >
> > - Lee & Ham, AZ-NAS: assembling zero-cost proxies for network architecture search, CVPR 2024
> >
> > In this work, the authors directly ensemble ZCPs by aggregating them non-linearly and use them as the fitness score for a genetic framework. While we separate ZCPs into different roles instead of straightforward aggregation, this work is very close to our own hence we added them to our related works section.
> >
> > - Lin & Luo, Per-architecture training-free metric optimization for neural architecture search, NeurIPS 2025
> >
> > This work introduces PO-NAS, a genetic framework which trains a surrogate model with ZCPs as auxiliary metrics to select the training samples and dynamically supplement architecture encodings, forming a hybrid surrogate+ZCP method. This work makes use of ZCPs as part of a predictor training pipeline, which is quite remotely connected to our work. Therefore we do not consider it to be a related work but included it as an example of recent effective methods in our introduction.
> >
> > - Lukasik et al., An evaluation of zero-cost proxies - from neural architecture performance prediction to model robustness, IJCV 2025
> >
> > This work is an analysis of the predictive performances of ZCPs. One of its key insights is the need to consider ZCPs jointly to evaluate challenging objectives such as robustness. Since this directly supports the usage of multiple ZCPs as in our work, we include it in our related works section.
> >
> > - Qin et al., Transferrable Surrogates in Expressive Neural Architecture Search Spaces, AutoML 2025
> >
> > This work uses ZCP and graph properties as lesser-fidelity representations to train surrogate models for the exploration of high expressivity search spaces built from CFGs (e.g einspace). Since this is also linked to ensembling ZCPs, we include it in our related works section.

---

> > > ### Author Response · Authors · 2026-03-31
> > > **Overhaul of the Related Works section**
> > >
> > > - Tybl and Neumann, Training-free neural architecture search through variance of knowledge of deep network weights, CVPR 2025
> > >
> > > This work introduces a novel ZCP based on Fisher Information theory. While this method displays strong correlation results, it did not report any search results from a full zero-cost pipeline. Since this metric was not present in our original experiments it would be out of place to include it in our paper. However, we note that it should definitely be studied in future zero-cost works. Furthermore, since the introduced metric is clearly not data-agnostic, its existence does not change the conclusions of our work.
> > >
> > > - White et al., How powerful are performance predictors in neural architecture search?, NeurIPS 2021
> > >
> > > This work evaluated all NAS paradigms together to determine the best predictor to use depending on the situation. Since the roster of ZCPs included is quite small and outdated compared to the current state-of-the-art, we do not feel the need to include it. As the reviewers aptly noted, our paper already references a lot of older works hence adding additional outdated benchmarks would be counter-productive.
> > >
> > > Additionally, we conducted our own survey of recent papers in the field, but did not find any of better relevancy than the ones suggested by the reviewer.

---

> > ### Author Response · Authors · 2026-03-31
> > **Paper revision**
> >
> > We have also applied other suggestions made by the reviewer:
> > - All wrong citations have been corrected
> > - We clearly differentiated our work from those of other related papers, especially ZCP ensembling methods, by highlighting the fact that our method can be used jointly with ZCP ensembling.
> > - We made the novelty of our paper clearer in the Introduction and Methods sections by highlighting that one of the main goals of ParaDis is the construction of the space- and data-agnostic pipeline. This should also better motivate the choice of synflow as the main metric for the experiments.
> >
> > Once again, we thank the reviewer for their helpful suggestions which we believe have greatly helped to improve the paper.

---

### Review · Reviewer_sCvz · 2026-03-18

**Summary Of Contributions:**

This paper shows that the correlations between zero-cost proxies in neural architecture search (NAS) and actual performance decrease for a well-performing architecture set. Then, the authors propose a genetic algorithm-based method that leverages a zero-cost proxy and the adaptive parameter count as explorer and discriminator, respectively. The proposed method, named ParaDis, is evaluated using NAS benchmark datasets and the DARTS search space. The experimental results show that ParaDis can find competitive architectures with existing zero-cost proxy-based methods.

**Audience:**

Yes

**Audience Explanation:**

The topic of this paper, improving zero-cost proxy-based NAS, is interesting and relevant to some TMLR's audience. Although some audience will be interested in the content of this paper, the advantages of the proposed ParaDis and the justification of the algorithm design should be clearly presented.

**Broader Impact Concerns:**

I do not have concerns about the ethical implications of this paper.

**Claims And Evidence:**

No

**Claims Explanation:**

The authors show that zero-cost proxies for the top 1% architectures in the search space are poorly correlated with ground-truth scores, a phenomenon they call the "top-rank gap." The reviewer thinks that this is not so surprising. The purpose of zero-cost proxies is to quickly filter out bad architectures. Therefore, it seems acceptable even if the top-rank gap exists, depending on the problem settings.

The design of the proposed ParaDis seems heuristic. The reason why the adaptive parameter count can be a good discriminator is not well explained. The authors should provide more justifications or theoretical evidence for the design of ParaDis.

Figure 6 indicates that the number of parameters is a strong competitor to the proposed ParaDis variants. It is unclear the advantage of ParaDis against the simple parameter count. The authors should provide more analysis on this point. Perhaps it might be better to conduct the statistical test.

Table 1 also shows that the performance of ParaDis is not significantly better than that of the baseline methods. For example, TE-NAS, MeCo, and SWAP-NAS still outperform ParaDis in terms of validation accuracy with reasonable computational time. Therefore, the advantage of ParaDis is not clear.

The aim of the ablation study in Section 5.3 is not clear. In the ablation study, the authors compare the performance of ParaDis with different selection schemes in the genetic algorithm. The motivation and insight from this ablation study are not well explained. The experimental result seems to indicate only a generic consequence, like "This highlights the importance of proper genetic algorithm design as seemingly small changes can be the difference maker between effective evolution and non-stochastic collapse." What is the unique observation of ParaDis in this ablation study?

**Requested Changes:**

- The current experimental result does not support the advantage of the proposed method over simple parameter count and other baselines. The authors should provide clear evidence of the effectiveness of ParaDis.

- The ablation study in Section 5.3 lacks clear motivation and unique insights. The authors should explain the important point and key insight from the ablation study.

- The authors exclude the NAS-Bench-101 in the experimental evaluation. The reason stated by the authors is that it does not provide significant insight, despite having large amounts of data. The reason is not convincing for the reviewer. The authors should explain why it cannot provide significant insight or present the result at least in the appendix.

---

> ### Author Response · Authors · 2026-03-27
>
> We thank the reviewer for their critical and constructive feedback. We address the concerns below.
>
> 1 - We respectfully disagree that top-rank gap can be considered an acceptable flaw for the following reasons:
> - The overarching goal of Neural Architecture Search from its inception is to aim to find the best possible architecture, especially in shallower search spaces such as the presented benchmarks. Whether the means are lower fidelity shouldn't change this goal.
> - ZCPs are routinely used for other purposes than search space shrinking i.e for efficient exploration of Pareto fronts [1], where correct ranking of high-performance architectures is a clear positive.
>
> 2 - On the topic of the effectiveness of ParaDis.
>
> The reviewer seems to have misinterpreted the aim of Figure 6. As we mention in the paper, the metric reported in Figure 6, which is average accuracy in the population, is important to ensure higher chances of culling out weaker architectures in the final stage of the search (i.e successive halving or any other scheme operating on a small number of candidates). Under the assumption that such algorithm can always find the best architecture in the population, the most relevant metric is the performance ceiling of the population, which we measure in Figure 5 via the maximum rank. We are open to feedback regarding better ways to word this piece of information and avoid misinterpretation.
>
> We acknowledge that given the numbers in Figure 6, the hypothesis test with hypothesis A: "ParaDis yields better average population accuracy than #params" will almost surely result in hypothesis A being rejected. However, this is not the aim of our method: bringing average population accuracy up to the levels of #params is actually a good result, since this unlocks the usability of most ZCPs.
>
> Furthermore, our experimental results in the DARTS space (Table 1) support the advantages of ParaDis compared to the #params baseline in a complex and competitive space. Additionally, compared to other methods, it removes priors on search space and dataset, which is a unique feature of our method ensuring its efficacy is transferrable across tasks. We will revise the paper to place greater emphasis on this advantage made possible by our framework.
>
> 3 - On the topic of the genetic setup ablation.
>
> This ablation is motivated by the lack of transparency of other papers concerning this point. Papers such as SWAP-NAS [2] report great results and theoretical insights on ZCP design but fail to describe the main intuitions behind genetic algorithm design, despite the modularity of the genetic framework. We argue that search methodology is equally as important as ZCP design.
>
> Since this could lead to better understanding of the behavior of NAS genetic algorithms for the community, we will keep investigating and report any new insights in a followup post. Our first intuition is that a certain level of stochasticity is required to break out of local population optima (i.e mutation loops) which our setups 1 and 2 fail to provide, hence the collapse to identical populations. Introducing tournament selection helps stochasticity, and pruning architectures globaly as in setup 3 speeds up exploration thus making it less likely to stabilize early. Since setup 4 with dual tournament selection does not improve performance significantly, we hypothesize that there are diminishing gains beyond a certain level of stochasticity.
>
> 4 - We softened our statement about NAS-Bench-101 since it was mainly motivated by computational budget constraints given the size of the space.
>
> References:
> [1] Javaheripi et al., LiteTransformerSearch: Training-free Neural Architecture Search for Efficient Language Models, NeurIPS 2022
> [2] Peng et al., SWAP-NAS: Sample-Wise Activation Patterns for Ultra-Fast NAS, ICLR 2024

---

### Author Response · Authors · 2026-03-31
**Summary of Paper Revisions**

We thank all the reviewers for their detailed feedback and detail below all the changes that were made to our submission:
- Some major changes were made to the Introduction and Methods section to clarify the key novelty of our work and the differences compared to closely related works. Specifically, we highlighted the space- and data-agnosticity aim of the full ParaDis pipeline and the unique explorer/discriminator framework, which is complementary to other ZCP ensembling methods.
- The Related Works section has been greatly enhanced with the inclusion of additional, more recent works suggested by reviewer WaJS. As a result, the position and novelty of the paper compared to the literature should be clearer.
- The source code for our ParaDis method has been added as supplementary material.
- All wrongly formatted citations have been fixed.
- The abstract has been modified to include empirical results
- The statement about the omission of NB101 in the benchmarks and experiments has been softened.
- We conducted additional benchmarking experiments with the Kendall-tau correlation metric, which we added to the appendix. The conclusions of this benchmark reinforce our original observation of the top-rank gap.
- We also ran benchmarks for the top-2% and top-5% cutoffs. These benchmarks show that, as expected, the top-rank gap worsens progressively as the quality of architectures increases.

---

### Decision · Action_Editor_Yc7Z · 2026-05-08

**Recommendation:** Accept as is

**Audience:**

Yes

**Audience Explanation:**

All reviewers acknowledged that the paper would be of interest to a part of the community.

**Claims And Evidence:**

Yes

**Claims Explanation:**

In their final recommendations, two reviewers supported acceptance while one recommended rejection. While all the reviewers stated that the novelty of the work was on the fence, the two positive reviewers acknowledged the interest of the analysis performed by the authors and of the insights arising from this analysis. They also commented positively on the authors' feedback and revisions made during the discussion period. Considering that novelty should not be taken as a criterion for TMLR and that the analysis and insights have been acknowledged by two reviewers, the AE believes that the submission is indeed sufficiently supported by convincing evidence.